# A hybrid semiconducting organosilica-based O$_2$ nanoeconomizer for on-demand synergistic photothermally boosted radiotherapy

Wei Tang[1], Zhen Yang[1✉], Liangcan He[1], Liming Deng[1], Parinaz Fathi [1], Shoujun Zhu[2], Ling Li[1], Bo Shen[3], Zhantong Wang[1], Orit Jacobson[1], Jibin Song [4], Jianhua Zou[1], Ping Hu[5], Min Wang[5], Jing Mu[1], Yaya Cheng[1], Yuanyuan Ma[1], Longguang Tang[1], Wenpei Fan [6✉] & Xiaoyuan Chen [7✉]

The outcome of radiotherapy is significantly restricted by tumor hypoxia. To overcome this obstacle, one prevalent solution is to increase intratumoral oxygen supply. However, its effectiveness is often limited by the high metabolic demand for O$_2$ by cancer cells. Herein, we develop a hybrid semiconducting organosilica-based O$_2$ nanoeconomizer pHPFON-NO/O$_2$ to combat tumor hypoxia. Our solution is twofold: first, the pHPFON-NO/O$_2$ interacts with the acidic tumor microenvironment to release NO for endogenous O$_2$ conservation; second, it releases O$_2$ in response to mild photothermal effect to enable exogenous O$_2$ infusion. Additionally, the photothermal effect can be increased to eradicate tumor residues with radioresistant properties due to other factors. This "reducing expenditure of O$_2$ and broadening sources" strategy significantly alleviates tumor hypoxia in multiple ways, greatly enhances the efficacy of radiotherapy both in vitro and in vivo, and demonstrates the synergy between on-demand temperature-controlled photothermal and oxygen-elevated radiotherapy for complete tumor response.

[1] Laboratory of Molecular Imaging and Nanomedicine, National Institute of Biomedical Imaging and Bioengineering, National Institutes of Health, Bethesda, MD 20892, USA. [2] State Key Laboratory of Supramolecular Structure and Materials, College of Chemistry, Jilin University, 130012 Changchun, P.R. China. [3] Institute of Radiation Medicine, Fudan University, 200032 Shanghai, P.R. China. [4] MOE Key Laboratory for Analytical Science of Food Safety and Biology, College of Chemistry, Fuzhou University, 350116 Fuzhou, P.R. China. [5] State Key Laboratory of High-Performance Ceramics and Superfine Microstructure, Shanghai Institute of Ceramics, Chinese Academy of Sciences, 200050 Shanghai, P.R. China. [6] State Key Laboratory of Natural Medicines and Jiangsu Key Laboratory of Drug Discovery for Metabolic Diseases, Center of Advanced Pharmaceuticals and Biomaterials, China Pharmaceutical University, 210009 Nanjing, P.R. China. [7] Yong Loo Lin School of Medicine and Faculty of Engineering, National University of Singapore, 117597 Singapore, Singapore. ✉email: beijinyz@126.com; wenpei.fan@cpu.edu.cn; chen.shawn@nus.edu.sg

A well-established hallmark of the tumor microenvironment (TME) is the rapid proliferation of cancer cells that need a large amount of oxygen and nutrients; however, an imbalance between oxygen supply and requirement leads to hypoxia in solid tumors[1,2]. Oxygen deficiency, in turn, challenges the therapeutic efficacy of most cancer treatments in the clinic, especially radiotherapy (RT)[3]. In RT, ionizing radiation damages cells by producing a radical on DNA (DNA•). According to the oxygen fixation hypothesis, the DNA radical can be further oxidized by molecular $O_2$ to generate DNA-OO•, thus inducing the damage fixation and DNA double-strand breaks. Of note, the radicals can be competitively reduced at the same time, especially under hypoxic condition, by thiol-containing compounds to restore the DNA to its original form, resulting in less DNA damage[2]. Various methods of modifying hypoxic radioresistance have been explored in clinical settings, such as increasing oxygen delivery through the blood with hyperbaric oxygen (HBO), normobaric oxygen/carbogen breathing, nicotinamide, blood transfusion, erythropoietin, or a combination of them; mimicking the oxygen effect on fixation of radiation-induced DNA damage in the radiochemical process with nitroimidazoles; destroying hypoxic cells, rather than sensitizing them, with hypoxic cytotoxins; and having a more direct radiation target in the cells with high linear energy transfer irradiation[4]. But hypoxic modification in the routine clinical practice remains inconclusive and very limited, partly because the above strategies are small and underpowered, or due to involvement of techniques that are difficult to be routinely practiced (e.g., HBO)[4]. Nowadays, with the advancement in nanotechnology, reshaping the hypoxic TME by providing extra oxygen sources with nanomaterials is emerging as an important alternative adjuvant therapeutic strategy to reverse oxygen-related tolerance in RT. Two major strategies have been explored. One is to use oxygen-generating nanomaterials, for example, manganese dioxide nanoparticles[5,6], gold nanoclusters[7], and catalase-encapsulated nanosystems[8,9], which can be transported to tumor sites to decompose hydrogen peroxide into oxygen. However, their outcomes are highly restricted by the local hydrogen peroxide concentration. The other is to employ oxygen-carrying nanoplatforms as oxygen shuttles for direct oxygen delivery[10,11]. For instance, we have developed semiconducting polymer-stabilized perfluorocarbon (PFC) nanodroplets for synergistic photothermal therapy (PTT) and oxygen self-sufficient photodynamic therapy with a single laser irradiation[12]. Yet, PFC nanodroplets are often associated with stringent experimental conditions and complications in co-encapsulation of other desirable therapeutic agents, due to the low boiling point and superhydrophobicity of PFCs. Therefore, solutions to stabilize the PFC entity in nanocarriers are highly desirable for efficient oxygen loading, controllable oxygen release, and facile drug co-delivery to alleviate tumor hypoxia by increasing intratumoral oxygen supply.

Although promising, the outcome of the above exogenous oxygen delivery strategies, or in other words "broadening sources" of $O_2$ strategies, is often greatly restricted by the high level of oxygen consumption in cancer cells due to cell respiration[13,14]. Despite that cancer cells acquire energy primarily through aerobic glycolytic metabolism, or the "Warburg effect", mitochondrial respiration is not diminished in cancer cells[15]. Instead, it plays an important role in tumor development and progression, which is evidenced by significantly elevated mitochondrial respiration rates in many cancer cell lines[16,17]. Unfortunately, very few studies have made efforts to design an anti-hypoxia nanodelivery system that aims to reduce expenditure of physiological oxygen by cancer cells. The concept of inhibiting oxygen consumption for hypoxia attenuation was first proposed by Secomb et al. in 1995[18]. Since then, several chemodrugs such as metformin[19–22],

phenformin[21], and atovaquone[23–25] have been applied to disturb mitochondrial respiration for improved tumor oxygenation. However, their side effects to normal cells cannot be neglected. Currently, nitric oxide (NO), a multifunctional signaling molecule involved in many physiological and pathological processes, has attracted increasing interest in not only normalizing tumor vessels for increased enhanced permeability and retention (EPR) effect[26–29], but also sensitizing cancer cells to other treatment modalities for synergistic cancer therapy[30]. For example, Howard-Flanders[31] demonstrated NO as an effective radiosensitizer as early as 1957 on hypoxic bacteria. It was proposed that the primary mechanism of NO-based radiosensitization was to fix radiation-induced DNA damage and mimic the oxygen effects on DNA legions[31]. Yet, it required a high level of NO concentration which may not be obtained in vivo due to its vasoactive complications. An alternative mechanism might be the interaction of NO with the oxygen-binding sites in mitochondria[32], leading to inhibition of cell respiration and conservation of physiological oxygen for sensitizing RT[33]. Of note, De Ridder et al.[34] first reported that NO can be endogenously generated through inducible isoform of NOS (iNOS) for radiosensitizaiton. On this basis, proinflammatory tumor infiltrates, for example, activated macrophages and T/NK-cells, can sensitize hypoxic tumors to RT through iNOS-dependent pathways by production of proinflammatory mediators and NO[35,36]. But the percentage of the tumor-associated immune cells varies in different tumor types and it may need to combine with immunostimulators for enough NO production. For tumor-specific exogenous NO delivery, many NO-releasing molecules, including S-nitrosothiols, L-arginine, and metal nitrosyls, have been reported to release NO in response to various endogenous/exogenous stimuli, such as acidity, enzyme, and ultrasound irradiation[30]. For instance, S-nitroso-N-acetyl-DL-penicillamine (SNAP) can be catalytically broken down to NO gas and disulfide in the presence of trace amounts of transition metal ions in biological environments (especially under acidic condition)[37]. However, their short half-lives, poor tumor targeting, and potential side effects usually imped the in vivo use. Notably, in comparison with the external stimuli-triggered burst-like NO release, prodrugs that respond to endogenous stimuli often induce a relatively slower and more sustained NO release, which may in turn more efficiently potentiate EPR enhancement and facilitate tumor accumulation of the circulating nanocarriers[29]. Several nanocarriers have been reported to incorporate NO-releasing molecules for radiosensitization, such as poly(lactide-co-glycolic)-block-poly(ethylene glycol) (PLGA-b-PEG) nanoparticles[38] and upconversion nanoparticle-engineered mesoporous silica core–shell structures[39]. But these carriers are either lack of diagnostic functionality or involved with potential premature drug release issues. More improvements can be made in the design of the nanocarriers. To this end, nanotheranostic platforms with effective incorporation and retention of PFC entities as well as TME-responsive NO prodrugs remain to be explored for highly efficient co-delivery of cell respiration inhibitors (e.g., NO) and exogenous oxygen resources (e.g., $O_2$) to overcome the restrictions of tumor hypoxia in RT.

In this work, we propose an $O_2$ nanoeconomizer for simultaneous $O_2$ conservation and infusion to relieve the intratumoral oxygen tension, based on an in situ polymerized, hollow-structured, and semiconducting polymer brush (SPB) and fluorocarbon (FC) chain co-hybridized organosilica nanoplatform (pHPFON) (Fig. 1a). First, the framework incorporation of the SPB moiety, with electron donor and acceptor backbone, grants the organosilica nanoplatform with not only excellent near-infrared (NIR) II fluorescence and photoacoustic (PA) contrast, but also efficient photothermal conversion, leading to an "all-in-

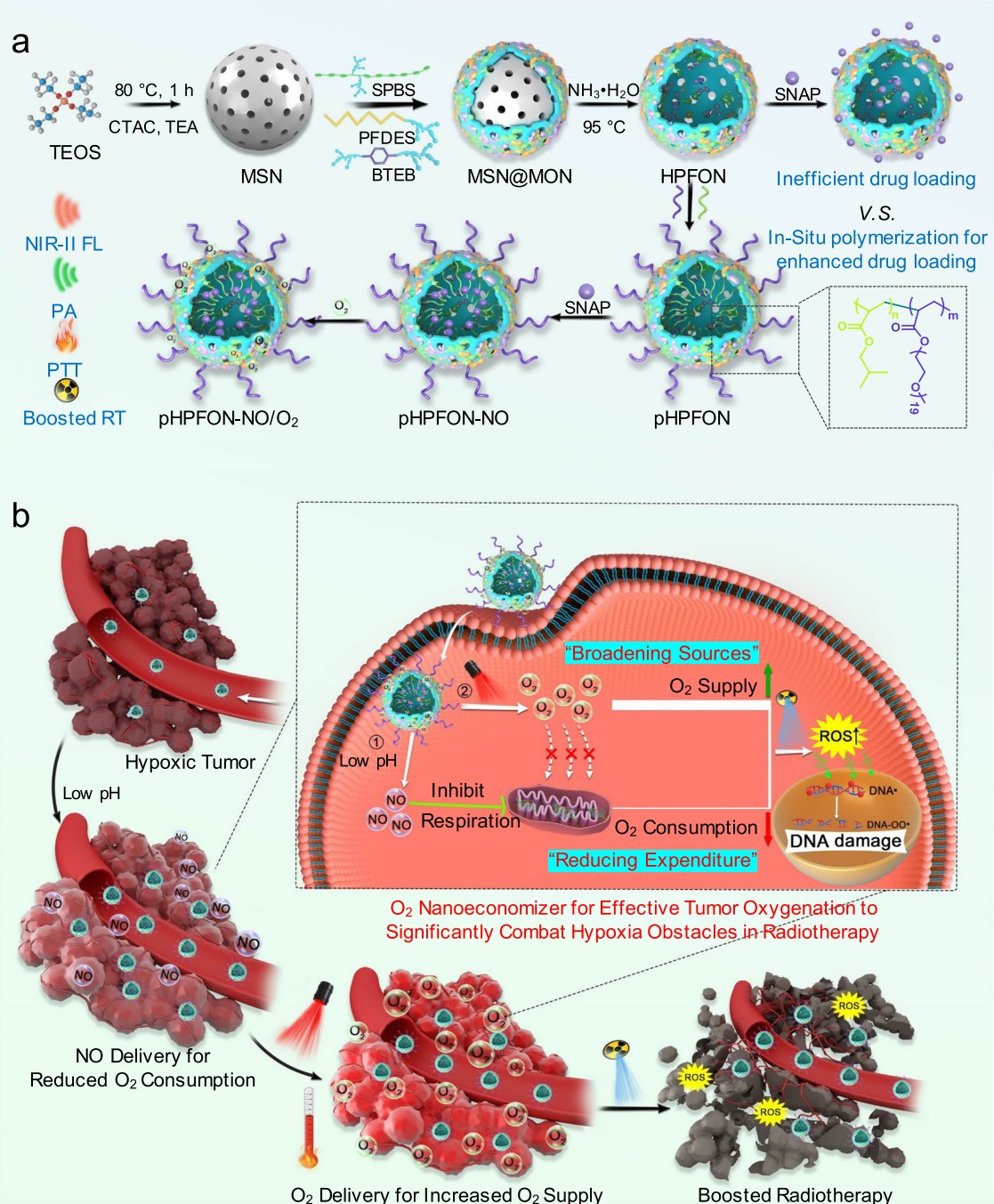

**Fig. 1 Schematic illustration of fabrication and boosted radiotherapy of the smart O$_2$ nanoeconomizer pHPFON-NO/O$_2$. a** Synthetic procedures. First, sub-50 nm semiconducting polymer brush/fluorocarbon/phenylene triple-hybridized HPFON was prepared by deposition of bissilylated organosilica precursors onto an MSN template via hydrolysis based on the chemical homology principle and selective MSN etching through an ammonia-assisted hot water etching strategy. Then, an in situ polymerization method was applied to conjugate alkyl chains and PEG polymers onto the inner and outer shell of the HPFON for enhanced hydrophobic drug loading as well as improved biocompatibility. Finally, SNAP and O$_2$ were loaded onto the resultant pHPFON to generate the pHPFON-NO/O$_2$. **b** Schematic illustration of the binary "reducing expenditure and broadening sources" tumor oxygenation strategy by programable delivery of NO and O$_2$ with pHPFON-NO/O$_2$ to overcome hypoxia-associated therapy resistance for boosted anti-cancer radiotherapy.

one" theranostic nanocarrier with both diagnostic and therapeutic functions. Second, the framework hybridization of the FC moiety avoids the disadvantages of superhydrophobicity and volatility of free PFC liquid, offering facile oxygen loading and photothermally controlled oxygen release ("broadening sources" of O$_2$). Third, the in situ polymerization of PEG and hydrophobic alkyl chains allows for improved biocompatibility and enhanced loading and retention of hydrophobic NO prodrugs in the cavity

of pHPFON to release NO gas for efficient blockage of intrinsic oxygen consumption ("reducing expenditure" of O$_2$). Fourth, pHPFON, with a size of around 50 nm, exhibits high tumor accumulation through EPR effect-based passive tumor targeting, in which the locally released NO can further improve the accumulation through tumor vasculature normalization. Fifth, the PTT effect of SPB can be precisely controlled to either a mild hyperthermia level for tumor oxygenation improvement and

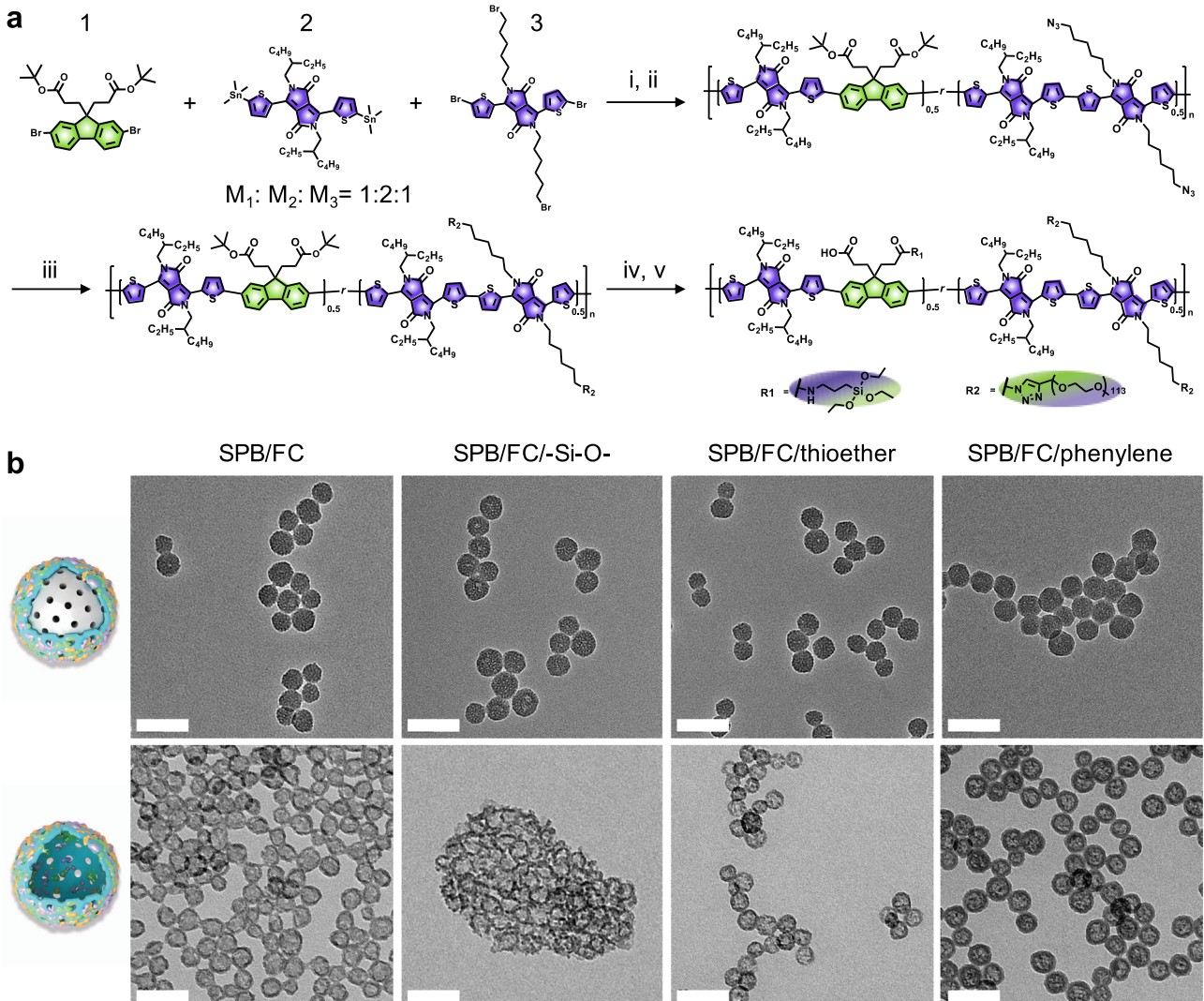

**Fig. 2 Morphology optimization of the pHPFON nanocarrier. a** Design and synthesis of the semiconducting polymer brush (SPB) silane precursor. (i) bis(triphenylphosphine)palladium(II) dichloride, 2,6-di-*tert*-butylphenol, toluene, 100 °C, 6 h; (ii) NaN₃, THF/DMF, 25 °C, 12 h. (iii) N,N,N',N", N'''-pentamethyldiethylenetriamine, mPEG₅₀₀₀-alkyne, CuBr, THF, 25 °C, 48 h. (iv) trifluoroacetic acid (TFA), 24 °C, 24 h. (v) (3-Aminopropyl) triethoxysilane (APTES), EDC/NHS, DMF, 25 °C, 12 h. **b** TEM images of different SPB and fluorocarbon (FC) chain co-hybridized formulations: SPB/FC, SPB/FC/-Si-O-, SPB/FC/thioether, or SPB/FC/phenylene-hybridized MSN@MON (top) and HPFON (bottom). Scale bar, 50 nm. Experiments were performed three times with similar results.

boosted RT or to a higher temperature to eliminate the residual tumor masses with radiation resistance due to the other factors. Collectively, the design of the SNAP and $O_2$ co-encapsulated pHPFON nanoplatform (pHPFON-NO/$O_2$) demonstrates programmatically acidity-activatable NO release and laser-activatable $O_2$ delivery, resulting in a smart $O_2$ nanoeconomizer to predominantly alleviate tumor hypoxia for on-demand synergistic PTT-boosted RT (Fig. 1b).

## Results

**Design and synthesis of pHPFON**. The framework hybridization strategy provides a simple and elegant method to functionalize hollow mesoporous organosilica nanoparticle (HMON) with desirable physiochemical merits[40]. To endow the nanocarriers with NIR-II fluorescence, PA contrast, and photothermal capacity, an NIR-II SPB-based silica (SPBS) precursor was synthesized (Fig. 2a). Briefly, the backbone of the SPBS was fabricated via the Stille coupling strategy with diketopyrrolopyrrole-based electron acceptor and fluorine-

based electron donor monomers. Then PEG (Mw, 5000) chains and 3-(triethoxysilyl)propylamine silica source were sequentially coupled to the electron donor and acceptor backbone segments by click reaction and esterification, respectively. $^1$H-NMR spectra confirmed the successful synthesis of the intermediates (Supplementary Fig. 1). Absorbance and fluorescence spectra verified the NIR-II properties of the resultant SPBS (Supplementary Fig. 2). In addition to the SPB, FC chains were incorporated into the framework of HMON for $O_2$ delivery. Based on the chemical homology principle, the framework hybridization of mesoporous organosilica nanoparticle (MON) was achieved by co-hydrolysis of a mixture of SPBS and 1H,1H,2H,2H-perfluorodecyltriethoxysilane (PFDES) onto an inorganic mesoporous silica nanoparticle (MSN) template. After selectively etching away the inorganic MSN core in the resultant MSN@MON with ammonia, a hollow-structured SPB/FC co-incorporated MON (HPFON) was obtained, in which the selectivity was attributed to the weaker alkaline etching resistance of the Si–O bond within the MSN core than the Si–C bond within the MON shell[41].

We further optimized the morphology of the resultant HPFON by introducing a third type of silica precursor, for example, tetraethyl orthosilicate (TEOS, with inorganic Si–O bonds), bis[3-(triethoxysilyl)-propyl]tetrasulfide (BTES, with thioether moieties), and 1,4-bis(triethoxysilyl)benzene (BTEB, with a phenylene moiety). As shown in the transmission electron microscopy (TEM) images in Fig. 2b, all MSN@MON formulations exhibited uniform and well-dispersed spherical structures of a similar size. After the ammonia etching, however, only the phenylene-hybridized HPFON retained a uniform spherical morphology with a hydrodynamic size of around 65 nm as determined by dynamic light scattering. The other formulations contained distorted and easy-to-aggregate hollow spheres with hydrodynamic sizes ranging from 116 to 390 nm (Fig. 2b and Supplementary Fig. 3). It is apparent that the phenylene incorporation enhanced the stiffness of HPFON. In addition, introduction of the phenylene group could significantly reduce the hemolytic effect of HMON by decreasing the amount of surface-exposed silanol[41,42]. The SPB/FC/phenylene multiple-hybridized HPFON exhibited stability in PBS without substantial size changes over a week (Supplementary Fig. 4). Therefore, we chose the SPB/FC/phenylene multiple-hybridized HPFON formulation for further use.

The framework hybridization was verified through detection of signals for Si, C, O, and F in elemental mapping analysis (Fig. 3a) and energy dispersive X-ray spectroscopy (EDS) spectrum (Supplementary Fig. 5). In comparison, F was not detected in elemental mapping or EDS analysis of SPB/phenylene dual-hybridized HMON, demonstrating that the framework incorporation strategy was under good control (Supplementary Fig. 6). The HPFON had a large surface area of 632.5 m$^2$/g (Fig. 3b) and a well-defined mesoporous structure with a pore size of around 3.5 nm (Fig. 3c). Notably, these pores on the HPFON not only provided channels for drug loading, but also offered opportunities for unintended leakage of the cargoes. To solve this general issue in mesoporous drug delivery systems, we further conjugated the HPFON with hydrophobic polymers via an in situ polymerization approach to facilitate encapsulation and retention of hydrophobic drugs via hydrophobic–hydrophobic interactions (Fig. 1a)[43]. Meanwhile, PEG chains were co-conjugated during the polymerization process to ensure excellent biocompatibility of the nanoplatform. The resultant pHPFON demonstrated a SNAP loading ratio of 4.74 wt% after three high-speed centrifugation and wash cycles, whereas that in HPFON without surface polymerization was only determined to be 0.63 wt% by comparing the absorbance of unloaded content with the SNAP absorbance standard curve (Supplementary Fig. 7).

### Characterization of the pHPFON nanoplatform.

Next, we examined the spectroscopic properties and responsive cargo release of the pHPFON. It exhibited a broad NIR absorption from 600 to 1000 nm, with a strong peak at around 808 nm (Fig. 3d), and a high NIR-II fluorescence emission ranging from 1000 to 1400 nm, which peaked at around 1100 nm (Fig. 3e). The pHPFON exhibited a positive concentration-dependent PA contrast and photothermal effect (Fig. 3f, g). At a concentration of 1 mg/mL, 808-nm laser irradiation (1 W/cm$^2$) induced a temperature increase of 17.5 °C in 1 min and 40.6 °C in 5 min. To investigate the pH-responsive NO release, SNAP-loaded pHPFON (pHPFON-NO) was incubated in PBS at various pH values (Fig. 3h). At pH 7.4, pHPFON-NO underwent slow SNAP decomposition, with less than 5% payload release in 12 h. In contrast, a steadily accelerated NO release was observed at lower pH. This was because of SNAP decomposition in biological environments with the involvement of H$^+$[37], implying efficient

NO delivery when the pHPFON-NO was internalized by acidic endosomes/lysosomes of cancer cells or penetrated the acidic TME. Moreover, the pHPFON could be used for oxygen delivery owing to the strong physical affinity between the hybridized FC chains and O$_2$ molecules. The O$_2$-saturated pHPFON (pHPFON-O$_2$) did not release O$_2$ in the dark. In contrast, upon laser irradiation, the O$_2$ release rapidly increased and plateaued within 5 min (Fig. 3i), suggesting sustained and sufficient O$_2$ release in response to the photothermal effect. These observations indicated that the pHPFON-NO/O$_2$ might be an ideal candidate for sustained NO delivery and on-demand O$_2$ release in tumors.

### In vitro programmable NO/O$_2$ release.

Before investigating the intracellular NO and O$_2$ release, we examined the cellular uptake and biocompatibility of pHPFON. Both confocal microscopy and flow cytometry were used to demonstrate the time-dependent cellular uptake of fluorescein isothiocyanate (FITC)-labeled pHPFON (Supplementary Fig. 8). Moreover, we confirmed that pHPFON and its cargo-encapsulated formulations (i.e., pHPFON-NO/O$_2$, pHPFON-NO, and pHPFON-O$_2$) showed negligible cytotoxicity against U87MG cells in the dark (Supplementary Fig. 9). The NO and O$_2$ deliveries of pHPFON-NO/O$_2$ were monitored with an NO probe 4-amino-5-methylamino-2′,7′-difluorofluorescein diacetate (DAF-FM, green fluorescence) and a hypoxia indicator [Ru(dpp)$_3$]Cl$_2$ (red fluorescence) in hypoxic U87MG cells under confocal microscopy (Fig. 4a and Supplementary Fig. 10). Strong green fluorescence was only observed in the pHPFON-NO/O$_2$ or pHPFON-NO-treated cells with or without laser irradiation, which was attributed to the acidity-responsive SNAP decomposition upon internalization into acidic cellular endosomes/lysosomes. As expected, significantly weakened red fluorescence was displayed in the pHPFON-NO/O$_2$ or pHPFON-O$_2$-treated cells after laser irradiation, due to the photothermal effect-triggered O$_2$ release. Interestingly, along with the increased green fluorescence, pHPFON-NO also moderately reduced the [Ru(dpp)$_3$]Cl$_2$ signals in both the irradiated and non-irradiated cells, implying that the NO release may also be able to reduce hypoxia levels. The same results were detected when analyzing the intracellular fluorescence intensity with flow cytometry (Fig. 4b), which undoubtedly confirmed the programable NO and O$_2$ delivery with our proposed O$_2$ nanoeconomizer pHPFON-NO/O$_2$.

### NO delivery as a "reducing expenditure" oxygenation strategy to boost RT.

NO can competitively bind to the O$_2$ binding sites of cytochrome c oxidase (CcO)[32], which is an important component in the conversion of oxygen to water in the cell respiration chain[44,45]. We therefore speculated that upon cellular uptake of pHPFON-NO, the spontaneously released NO would enable efficient inhibition of mitochondrial respiration, in turn facilitating RT through downregulation of tumor hypoxia (Fig. 4c). Fig. 4d illustrates the decline of CcO activities in hypoxic U87MG cells after 24 h of incubation with various concentrations of pHPFON-NO. In comparison with pHPFON and PBS, pHPFON-NO demonstrated a much stronger potency in CcO activity inhibition (Fig. 4e). As a result of the respiration chain blockage, pHPFON-NO depolarized the mitochondrial membrane potential by showing a substantially decreased ratio in red (JC-1 aggregates, high potential, healthy status) to green (JC-1 monomer, low potential) in a JC-1 assay (Fig. 4f and Supplementary Fig. 11). This NO-mediated mitochondrial dysfunction was also demonstrated by a much lower ATP content of 53.8% in cells treated with pHPFON-NO than the control groups (Fig. 4g). Remarkably, the oxygen consumption rate assay only detected around 25% intracellular oxygen decrease after being exposed to

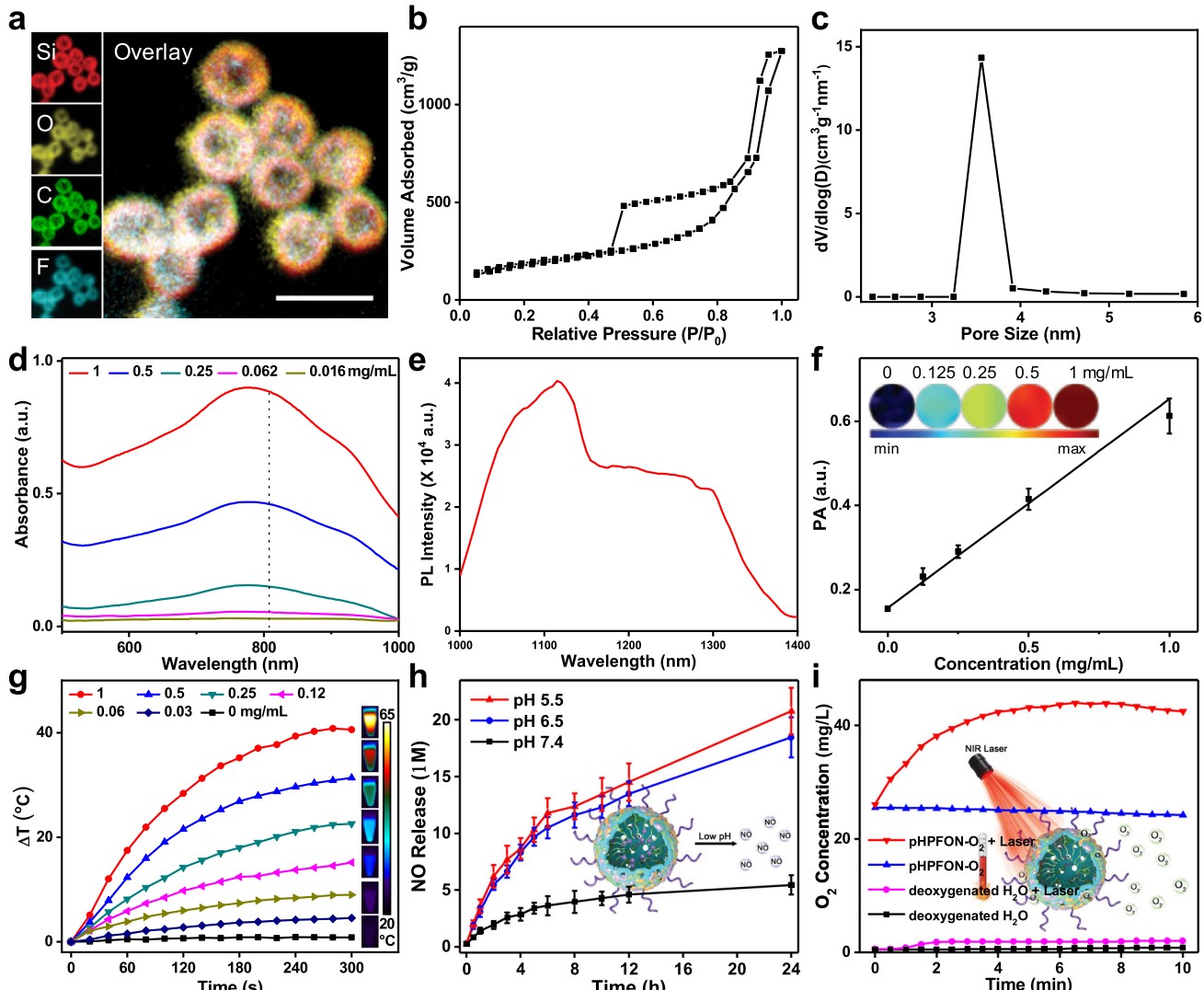

**Fig. 3 Characterization of pHPFON nanoplatform. a** Elemental mapping. Scale bar, 50 nm. **b** N2 adsorption–desorption isotherm and **c** corresponding pore-size distribution. **d** UV–vis spectra of pHPFON at various concentrations. **e** NIR-II fluorescence spectrum. **f** Plot of PA signal versus concentration. Inset: PA images at concentrations ranging from 0 to 1 mg/mL. **g** Photothermal heating curves and images of pHPFON at different concentration under 808-nm laser irradiation at 1 W/cm². **h** Cumulative NO release from pHPFON-NO at different pH values. Inset: schematic illustration of low-pH-induced NO release. **i** Cumulative O2 release from pHPFON-O2 in response to the laser irradiation. Inset: schematic illustration of laser-activatable O2 release. Experiments were performed twice (**b**–**e**, **g**, **i**) or three times (**a**, **f**, **h**), with similar results. Data are presented as mean ± s.d.

pHPFON-NO for 30 min, whereas those in pHPFON and PBS-treated groups were around 40% (Fig. 4h). This "reducing expenditure" oxygenation strategy further induced an apparent downregulation of hypoxia-inducible factor 1-alpha (HIF-1α, the hallmark of hypoxic cells) in the pHPFON-NO-treated hypoxic U87MG cells (Fig. 4i and Supplementary Fig. 12). Upon X-ray irradiation, we evaluated the reactive oxygen species (ROS) generation and DNA double-strand break with a fluorogenic probe, 2′,7′-dichlorodihydrofluorescein diacetate (H2DCFDA), and a phosphorylated protein biomarker, γ-H2Aχ, respectively. As expected, the pHPFON-NO-treated cells with X-ray irradiation exhibited the highest level of 2',7'-dichlorofluorescein (DCF) fluorescence (Supplementary Fig. 13) and positive anti-γ-H2Aχ staining (Supplementary Fig. 14) compared with other groups, which was attributed to the NO-mediated hypoxia relief.

**O2 delivery as a "broadening sources" oxygenation strategy to boost RT.** The radiosensitization effect from O2 delivery was examined with pHPFON-O2 to exclude the impact from NO. We hypothesized that the laser-activatable O2 release of pHPFON-O2 could effectively increase intracellular oxygen concentration, downregulate HIF-1α expression, elevate X-ray-mediated ROS generation, augment DNA damage, and finally lead to cancer cell death (Fig. 4j). Anti-HIF-1α immunofluorescence staining demonstrated a predominantly weakened green fluorescence in the pHPFON-O2 + Laser treatment group, while those of other groups without laser irradiation or O2 pre-saturation were almost the same as the PBS control. This confirms the photothermal effect-induced cellular oxygenation with pHPFON-O2 (Fig. 4k and Supplementary Fig. 15). Notably, this "broadening sources" oxygenation process further enhanced ROS production and oxidative DNA damage in RT (Fig. 4l), showing a much stronger positive DCF and anti-γ-H2Aχ staining level in the O2-sensitized RT than other non-O2-supplying RT treatments. In addition, negligible positive fluorescent signals were detected in the cells treated with different nanoparticles but without receiving X-ray irradiation (Supplementary Figs. 16 and 17). These observations clearly demonstrated that the pHPFON could act as an efficient

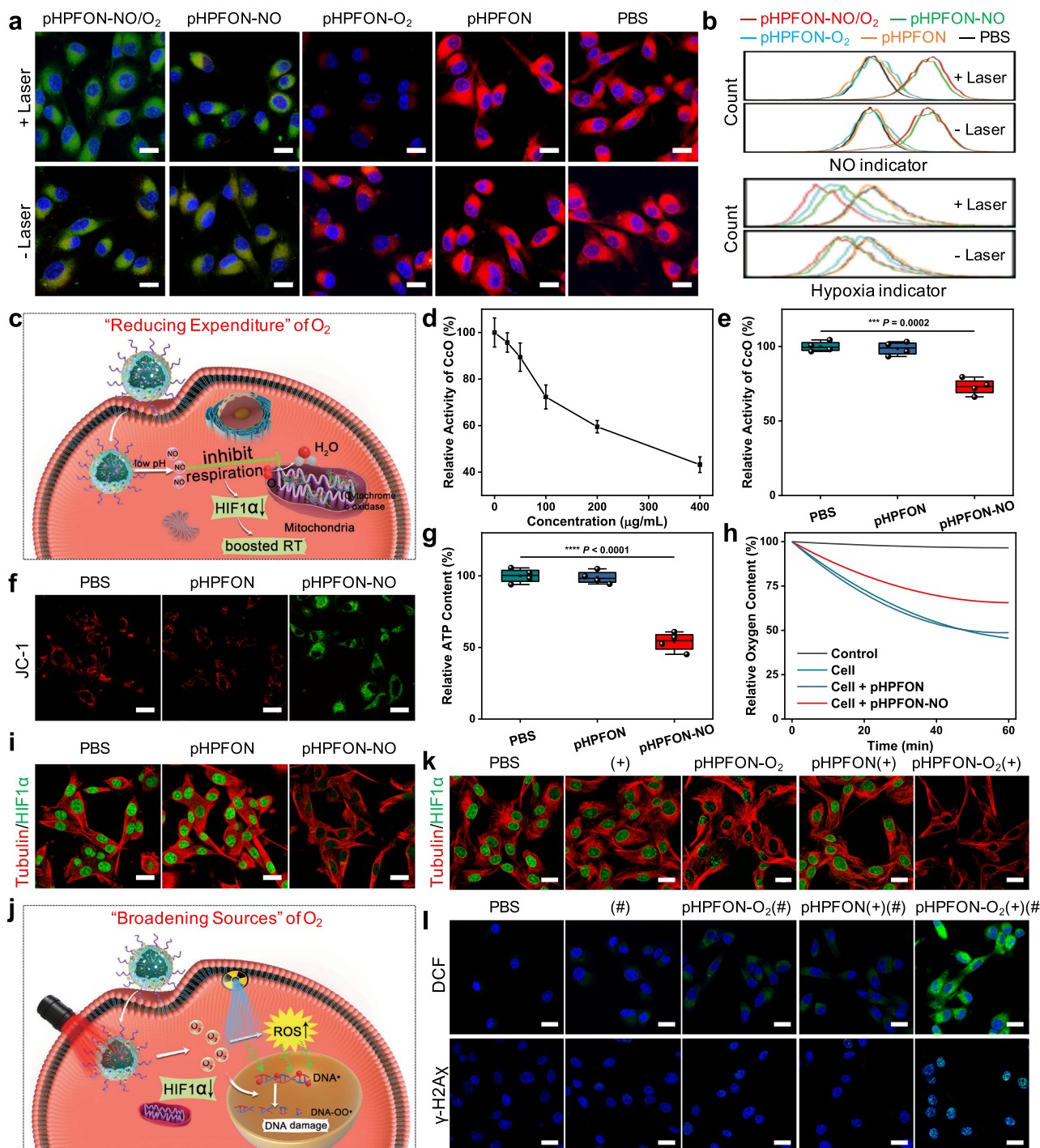

O$_2$ shuttle to transport oxygen to hypoxic cells and boost oxidative DNA damage in RT. Therefore, we expected significantly advanced RT efficacy with pHPFON-NO/O$_2$ through the binary O$_2$-economic tactic: on the one hand, the spontaneous NO release upon cellular uptake reduced the physiological O$_2$ consumption, on the other hand, the laser-activatable O$_2$ delivery supplied the cells with exogeneous oxygen.

**In vitro NO and/or O$_2$ sensitized RT**. The effective hypoxia alleviation by NO and O$_2$ delivery potentiated RT treatment. As shown in MTT assays in Fig. 5a, b, an X-ray dose-dependent RT effect was observed when treating U87MG cells with O$_2$ (pHPFON-O$_2$ + Laser), NO (pHPFON-NO), or sequential NO/

O$_2$ (pHPFON-NO/O$_2$ + Laser) delivery under both hypoxic and normoxic conditions. The NO/O$_2$ co-delivery demonstrated the most effective cell destruction of all groups under the same X-ray irradiation dose. In addition, cells treated with pHPFON plus laser or pHPFON alone showed negligible add-on effects in RT, therefore excluding the possibility of the mild photothermal effect or pHPFON nanocarrier-mediated RT enhancement. To further investigate the therapeutic effect of pHPFON-NO/O$_2$, we examined the survival fractions of U87MG cells by the colony formation assay after different treatments under normoxic and hypoxic conditions (Fig. 5c). The NO and/or O$_2$ sensitized RT treatments (groups g1–g3) significantly damaged cancer cells by greatly reducing their survival under both conditions. In contrast, without the delivery of NO or O$_2$, the RT treatments led to

**Fig. 4 In vitro programmable release and radiosensitizing effect of NO and $O_2$. a** Confocal images of hypoxic U87MG cells treated with different pHPFON formulations for 24 h, with or without subsequent 808-nm laser irradiation (1 W/cm$^2$, 3 min). Green, DAF-FM (4-amino-5-methylamino-2′,7′-difluorofluorescein, NO indicator). Red, [Ru(dpp)$_3$]Cl$_2$ (hypoxia indicator). Blue, DAPI. Scale bar, 20 μm. Experiments were performed three times with similar results. **b** Flow cytometry analysis of hypoxic U87MG cells receiving the same treatments in **a**. Experiments were performed twice with similar results. **c** Schematic illustration of the NO delivery-based "reducing expenditure" oxygenation strategy for boosted RT. The low-pH-induced NO release would inhibit mitochondrial respiration, downregulate HIF-1α expression, and boost RT efficacy. **d** Relative activity of cytochrome c oxidase (CcO) after incubating hypoxic U87MG cells with pHPFON-NO at different concentrations for 24 h. **e–i** Effect of cell respiration inhibition by the pHPFON-NO. **e** Relative CcO activity, **f** JC-1 assay (green, JC-1 monomer. Red, J-aggregates. Scale bar, 20 μm), **g** relative ATP contents, **h** oxygen consumption capacity, and **i** HIF-1α expression (green, HIF-1α; red, tubulin. Scale bar, 20 μm) after co-incubation of hypoxic U87MG cells with pHPFON-NO, pHPFON, or PBS overnight. **j** Schematic illustration of the $O_2$ delivery-based "broadening sources" oxygenation strategy for advanced RT. The laser-activatable $O_2$ release would downregulate HIF-1α expression and augment X-ray-induced oxidative DNA damage. **k** Anti-HIF-1α staining in hypoxic U87MG cells after different treatments. Green, HIF-1α; red, tubulin. Scale bar, 20 μm. **l** Evaluation of intracellular ROS generation and DNA damage with H$_2$DCFDA assay and anti-γ-H2Aχ staining after different treatments. Green, 2′,7′-dichlorofluorescein (DCF) or γ-H2Aχ; blue, DAPI. Scale bar, 20 μm. For **k**, **l**, (+) stands for 808-nm laser irradiation at 1 W/cm$^2$ for 3 min applied after 24 h of incubation with nanoparticles; (#) stands for 4-Gy X-ray irradiation following the laser irradiation, if applicable. n = 4 biologically independent samples per group (**d**, **e**, **g**). Experiments were performed three times with similar results (**f**, **h**, **i**, **k**, **l**). Data are presented as mean ± s.d. in **d**. For the boxplots, the middle line is the median, the lower and upper hinges correspond to the first and third quartiles, and whiskers represent ±1.5 interquartile range (**e**, **g**). Two-tailed Student's t-test. ***P < 0.001. ****P < 0.0001.

significantly lower cell proliferation in normoxia than in hypoxia (groups g4–g6), which unambiguously confirmed that pHPFON-NO/$O_2$ could effectively overcome the hypoxia barrier and remarkably potentiate the RT treatment. Of note, dose-dependent RT effect was also observed in the NO and/or $O_2$ sensitized cells by the colony formation assay (Supplementary Fig. 18). Furthermore, we assessed the RT-induced oxidative damage to DNA in hypoxic U87MG cells using the comet assay (Fig. 5d), in which the damaged DNA displayed a long tail of fluorescent stain. A much longer tail DNA stain was shown in NO/$O_2$-dual-sensitized RT (oxyRT), NO-sensitized RT, and $O_2$-sensitized RT groups than that in the X-ray alone or PBS alone group, with the ratio of tail DNA to head DNA determined to be 1.36, 0.54, 0.60, 0.13, and 0.02, respectively (Fig. 5e).

**In vitro synergistic photothermal and RT.** MTT assays in Fig. 5f revealed greatly enhanced cytotoxicity in the combination group (group T1) than PTT (group T2) or oxyRT (group T3) monotherapy, showing a fraction of viable cells (f) of 20.4 ± 2.4, 38.8 ± 3.6, and 68.3 ± 2.7%, respectively. The synergy between PTT and oxyRT was verified by the calculation of $f_{additive} = (f_{PTT} \times f_{oxyRT})$ > $f_{PTT+oxyRT}$[46–48]. In addition, negligible cell death was observed in the pHPFON-NO/$O_2$ plus low-dose laser irradiation (group T4) and the X-ray irradiation alone (group T5) groups, thus excluding the potential cell damage from the NO and $O_2$ delivery or the mild hypothermia effect. The results also verified essential RT sensitization from the programable NO and $O_2$ delivery. The same conclusion could be drawn from the Calcein-AM and ethidium homodimer-1 (EthD-1) co-staining results (Fig. 5g and Supplementary Fig. 19a) and the Annexin V-FITC/propidium iodide (PI) flow cytometry analysis (Fig. 5h, i and Supplementary Fig. 19b).

**In vivo multi-modal imaging of pHPFON-NO/$O_2$.** To find the best time window for the RT irradiation, we intravenously injected $^{64}$Cu-labeled or nonradioactive pHPFON-NO/$O_2$ into U87MG tumor-bearing mice and monitored their migration and tumor accumulation with positron emission tomography (PET) or NIR-II imaging. PET images (Fig. 6a) indicated gradual tumor accumulation with a peak of 9.80 ± 0.954% ID/g at 24 h post-injection (p.i.), which was relatively higher than those of other previously reported sub-50 nm HMONs (Fig. 6b)[41]. The accumulation enhancement might be attributed to the in situ NO release, which could promote tumor vasodilation and enhance the EPR effect[26]. Meanwhile, the liver and spleen uptake decreased

over time, indicating that the nanoparticles were successfully eliminated from the mice (Fig. 6b and Supplementary Fig. 20). The ex vivo radioactivities in major organs were consistent with the in vivo imaging results (Supplementary Fig. 21). Moreover, the NIR-II imaging displayed the same tendency with the PET results (Fig. 6c and Supplementary Fig. 22), exhibiting a relative tumor fluorescence signal of 1 ± 0.094, 2.75 ± 0.18, 3.88 ± 0.53, 3.27 ± 0.31 at 1, 4, 24, and 48 h p.i., respectively (Fig. 6d). To investigate the NO-mediated EPR enhancement, we meticulously compared the tumor accumulation of pHPFON-NO/$O_2$ and pHPFON with tumor PA imaging (Fig. 6e, f and Supplementary Fig. 23). Both nanoparticles showed the highest tumor uptake at 24 h p.i. and shared similar accumulation tendency. pHPFON-NO/$O_2$ demonstrated 1.31, 1.27, and 1.19 times higher tumor accumulation than pHPFON at 4, 24, and 48 h p.i., respectively, which confirmed the enhanced EPR effect with pHPFON-NO/$O_2$. For further therapy studies, we chose the 24 h p.i. time point to apply RT for maximum efficacy.

**In vivo treatment efficacy of pHPFON-NO/$O_2$.** Next, we moved to the in vivo therapeutic evaluation of pHPFON-NO/$O_2$ with U87MG tumor model. With 808-nm laser irradiation (1.0 W/cm$^2$), the tumor area of the pHPFON-NO/$O_2$-treated mice rapidly heated to around 44 °C within 3 min, and further increased to 48 °C at the end of 5 min. These two temperature conditions could induce effective $O_2$ release alone, or $O_2$ release plus photoablation, respectively. In contrast, the PBS-treated tumors maintained a temperature of below 36.2 °C during the irradiation (Fig. 7a, b). We also investigated the tumor oxygenation with pHPFON-NO/$O_2$ by visualizing the intratumoral hemoglobin oxygen saturation (s$O_2$) level with PA imaging (Fig. 7c). The relative s$O_2$ level steadily increased by 3.02-fold at 24 h p.i. of pHPFON-NO/$O_2$ and further explosively raised to 6.62-fold with a subsequent laser irradiation (Fig. 7d). These observations were consistent with the intratumoral HIF-1α expression (Fig. 7e, f), validating the "saving expenditure of $O_2$ and broadening sources" effect corresponding to the sequential NO and $O_2$ delivery, respectively. Undoubtedly, the remarkable PTT effect and tumor oxygenation enhancement with pHPFON-NO/$O_2$ would ensure potent tumor inhibition.

Therapy studies were performed on the U87MG tumor model (Fig. 7g). Complete tumor regression was only achieved by the combinational PTT and oxyRT treatment (Group T1). In contrast, the single modal PTT (Group T2) or oxyRT (Group T3) demonstrated a tumor growth inhibition rate of 87.4 and 70.8% at the 18th day after the treatments, suggesting a

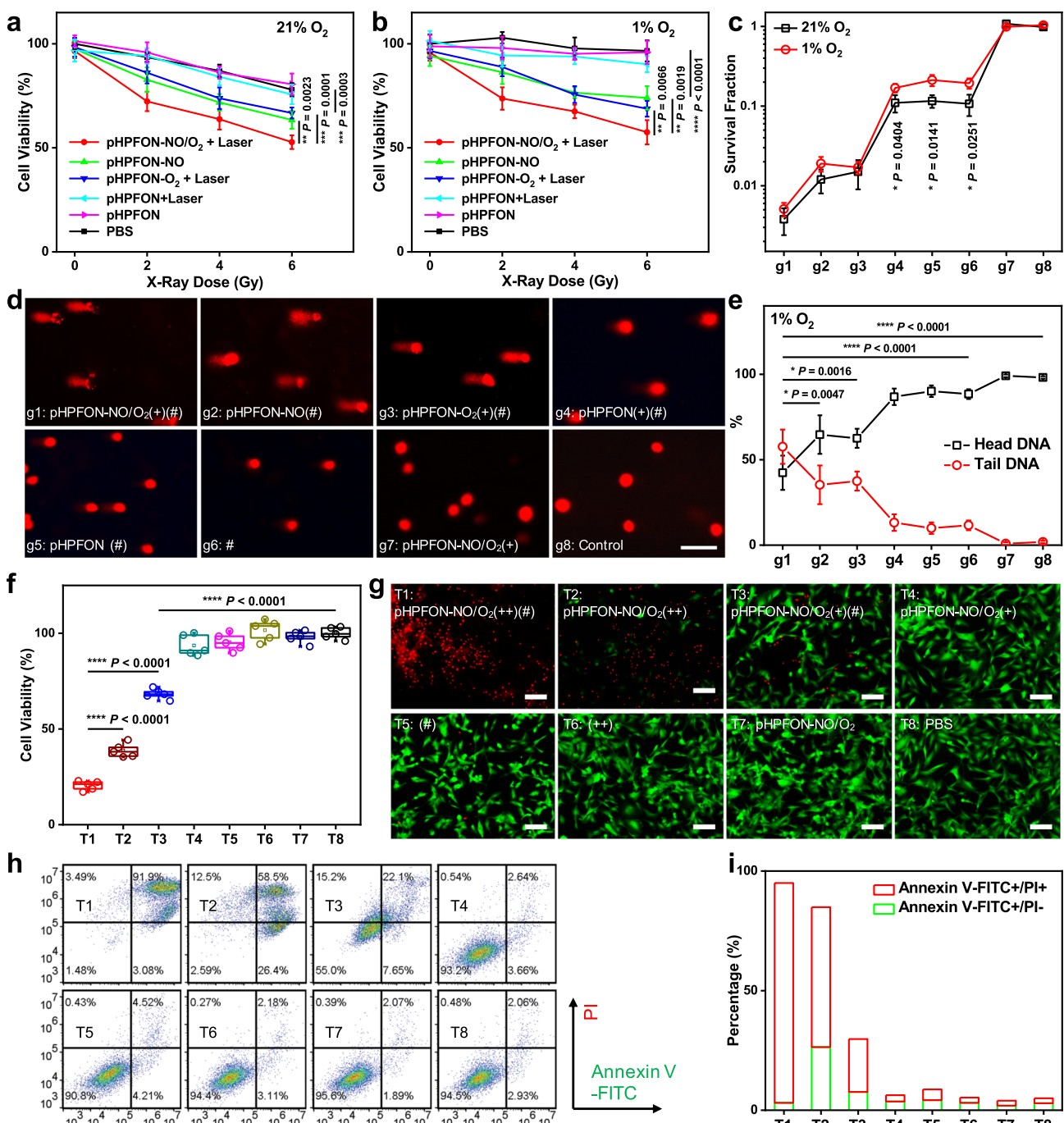

**Fig. 5 In vitro radiotherapy. a–e** NO and/or $O_2$-boosted radiotherapy. **a, b** Cell viabilities of **a** normoxic (21% $O_2$) and **b** hypoxic (1% $O_2$) U87MG cells subjected to different nanoparticle treatments, following by an X-ray irradiation at various doses (0, 2, 4, 6 Gy). In the groups with laser irradiation, the laser (808 nm) was applied after 24 h of incubation at a dosage of 1 W/cm$^2$ for 3 min. $n = 5$ biologically independent samples per group. **c** Survival fraction determined by colony formation assays in both normoxic and hypoxic U87MG cells after different treatments. $n = 3$ biologically independent samples per group. **d** Fluorescent DNA-stained images by comet assays in hypoxic U87MG cells after different treatments. Scale bar, 50 μm. Experiments were performed three times with similar results. **e** Quantification of DNA damage ($n = 6$ independent experiments) according to the images in **d**. For **c–e**, groups g1–g8: g1, pHPFON-NO/$O_2$ + Laser + X-ray; g2, pHPFON-NO + X-ray; g3, pHPFON-$O_2$ + Laser + X-ray; g4, pHPFON + Laser + X-ray; g5, pHPFON + X-ray; g6, X-ray; g7, pHPFON-NO/$O_2$ + Laser; g8, PBS. Laser (808 nm) was applied after 24 h of incubation with nanoparticles at 1 W/cm$^2$ for 3 min. X-ray was applied after the laser irradiation at a dose of 4 Gy. **f–i** In vitro synergistic photothermal and radiotherapy. **f** MTT assays ($n = 5$ biologically independent samples). **g** Live and dead assays ($n = 3$ biologically independent samples, with similar results). Green, Calcein-AM, live cells. Red, ethidium homodimer-1 (Eth-1), dead cells. Scale bar, 100 μm. **h** Flow cytometry analysis ($n = 2$ independent experiments, with similar results). **i** Quantitative analysis according to **h** on cells after different treatments. For **f–i**, groups T1–T8: T1, pHPFON-NO/$O_2$(++)(#); T2, pHPFON-NO/$O_2$(++); T3, pHPFON-NO/$O_2$(+)(#); T4, pHPFON-NO/$O_2$(+); T5, (#); T6, (++); T7, pHPFON-NO/$O_2$; and T8, PBS. (++) stands for 808-nm laser irradiation at 1 W/cm$^2$ for 5 min. (+) stands for 808-nm laser irradiation at 1 W/cm$^2$ for 3 min. (#) stands for a 4-Gy X-ray irradiation. Laser was applied after 24 h of incubation with nanoparticles and X-ray was applied after the laser irradiation, if applicable. Data are presented as mean ± s.d. Two-tailed Student's t-test. *P < 0.05. **P < 0.01. ***P < 0.001. ****P < 0.0001.

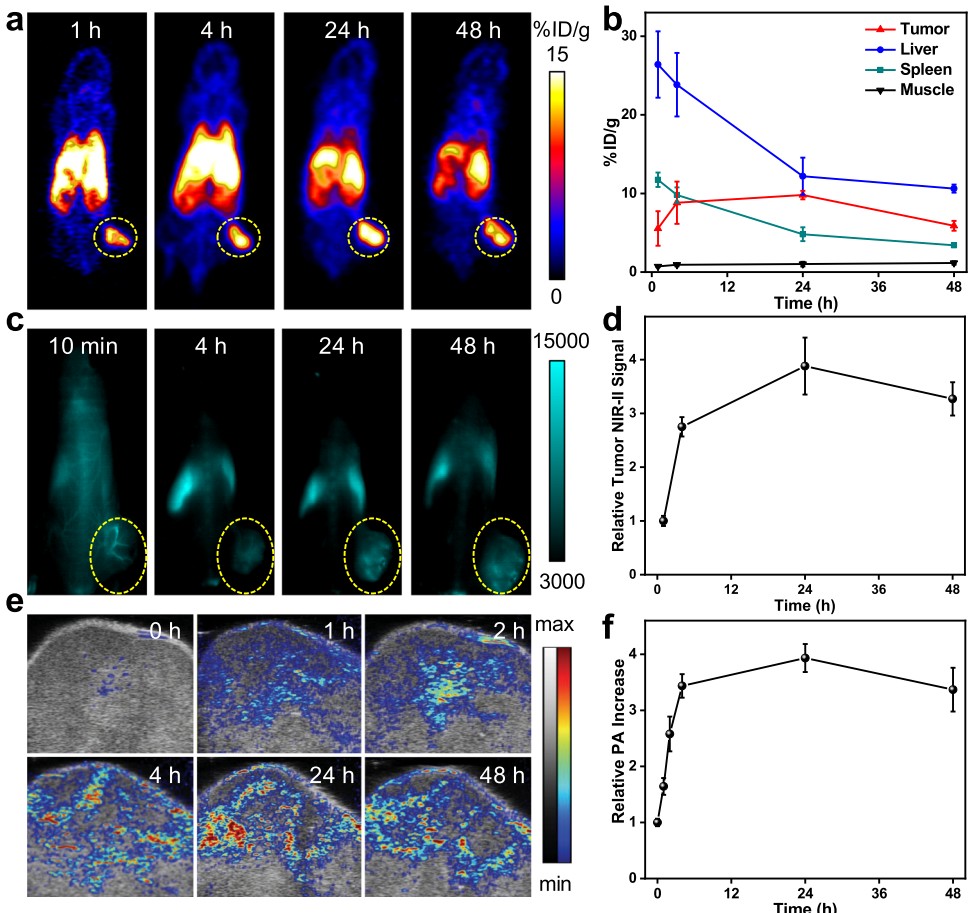

**Fig. 6 In vivo multi-modal imaging.** U87MG tumor-bearing mice were intravenously injected with $^{64}Cu$-labeled pHPFON-NO/$O_2$ for PET imaging or non-labeled pHPFON-NO/$O_2$ for NIR-II fluorescence and PA imaging. **a** Representative PET images at 1, 4, 24, 48 h p.i. Tumors are circled with yellow dots. **b** PET quantification on tumors and selected organs according to **a**. **c** Representative NIR-II images at selected time points. Tumors are circled with yellow dots. **d** Relative NIR-II signals at the tumor regions. **e** Representative tumoral PA images. **f** Relative PA signal changes. $n = 3$ biologically independent animals. Data are presented as mean ± s.d.

substantial cooperative enhancement effect between PTT and oxyRT (Fig. 7h). The X-ray irradiation alone (Group T5) only showed a moderate tumor inhibition rate of 55.9% at the 18th day, in which the attenuated RT efficacy was mainly due to the existence of hypoxic regions in the tumors. Therefore, the sensitization enhancement ratio of the combination group was calculated to be 1.41 and 1.79 over the oxyRT and RT treatment, respectively.

To further confirm the anti-cancer effect, tumors were collected after different treatments for histological analysis. Anti-$\gamma$-H2A$\chi$ staining analysis (Supplementary Fig. 24) showed much higher fluorescence signals in PTT/oxyRT-treated tumors than oxyRT or RT-treated ones. Hematoxylin and eosin (H&E) staining and TUNEL assays (Fig. 7i and Supplementary Fig. 25) revealed elevated levels of apoptotic and necrotic cells in the RT, oxyRT, PTT, and combinational PTT/oxyRT groups, but little cell damage in the other control groups, which reflected the observed tumor growth curves. In addition, to investigate whether hypoxia alleviation was a key player in the RT sensitization, tumors treated with oxyRT, NO-sensitized RT, $O_2$-senstized RT, or PBS were subjected to anti-HIF-1$\alpha$ (Supplementary Fig. 26) as well as anti-$\gamma$-H2A$\chi$ analysis (Supplementary Fig. 27). The oxyRT treatment with programmable NO and $O_2$ release demonstrated the lowest level of positive HIF-1$\alpha$ signals whereas the strongest positive $\gamma$-H2A$\chi$ fluorescence comparing with that in groups with either mere NO or $O_2$ release, verifying tumor hypoxia

attenuation is an important promoter in RT therapy. Despite efficient killing of cancer cells, the mouse body weight did not significantly drop during the treatment (Supplementary Fig. 28). Additionally, no apparent pathological abnormalities were detected in major organs from H&E staining (Supplementary Fig. 29), indicating minimal adverse effects.

Furthermore, the in vivo biosafety of pHPFON-NO/$O_2$ was evaluated. Healthy balb/c mice were intravenously injected with pHPFON-NO/$O_2$ and sacrificed at day 7 or day 14 p.i. for blood analysis. Complete blood count parameters showed little difference from those of the PBS-treated mice (Supplementary Fig. 30), suggesting no significant inflammation or infection caused by pHPFON-NO/$O_2$. Liver functional indexes, kidney functional biomarkers, and other examined metabolic parameters all remained in the normal ranges after the pHPFON-NO/$O_2$ administration (Supplementary Fig. 31), further verifying its negligible toxicity. Mouse body weight also steadily increased during the 14-day evaluation (Supplementary Fig. 32). H&E staining on major organs revealed no discernible acute pathological changes (Supplementary Fig. 33). Altogether, these results demonstrated the high biocompatibility of pHPFON-NO/$O_2$, suggesting its excellent biosafety for further clinical translation.

## Discussion
Combinational cancer therapy can make full use of the merits, and compensate for the intrinsic shortages, of each modality, thus

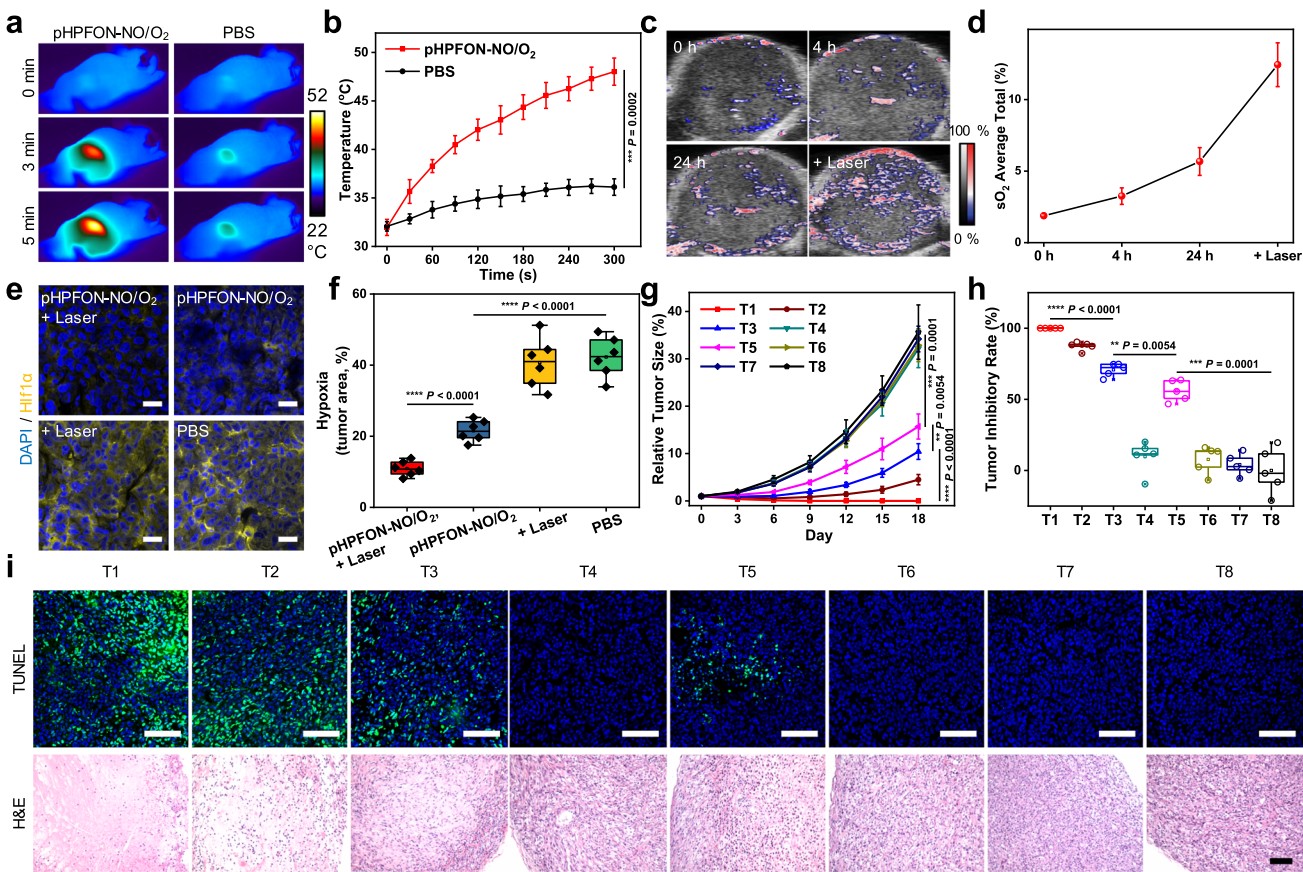

**Fig. 7 Therapy studies of pHPFON-NO/O$_2$ in a U87MG tumor model. a** Thermographic images of pHPFON-NO/O$_2$ or PBS-treated mice upon laser irradiation (808 nm, 1 W/cm$^2$) at 24 h p.i. at the tumor sites. $n = 3$ biologically independent animals. **b** Plots of tumor temperature with irradiation duration based on **a**. **c** PA imaging of tumor oxygenation change after pHPFON-NO/O$_2$ injection and a following laser irradiation at 24 h p.i. $n = 3$ biologically independent animals. **d** Quantification of oxygen saturation level according to **c**. **e** Anti-HIF-1α staining on tumors after different treatments. Laser was applied at 24 h p.i. Experiments were performed three times with similar results. Blue, DAPI; yellow, HIF-1α. Scale bar, 20 μm. **f** Quantitative tumoral hypoxia evaluation ($n = 6$ independent experiments) according to the results in **e**. **g** Tumor growth curves, **h** tumor inhibitory rates, and **i** tumor histological analysis with TUNEL (green, TUNEL; blue, DAPI.) and H&E staining assays after different treatments. $n = 5$ biologically independent animals with similar results. Scale bar, 100 μm. For **g–i**, groups T1–T8: T1, pHPFON-NO/O$_2$ + 5-min laser + X-ray; T2, pHPFON-NO/O$_2$ + 5-min laser; T3, pHPFON-NO/O$_2$ + 3-min laser + X-ray; T4, pHPFON-NO/O$_2$ + 3-min laser; T5, X-ray; T6, 5-min laser; T7, pHPFON-NO/O$_2$; and T8, PBS. The laser irradiation (808-nm, 1 W/cm$^2$) was applied at 24 h p.i. The X-ray irradiation was at a dosage of 8 Gy and applied at 24 p.i. following laser irradiation (if applicable). Data are presented as mean ± s.d. (**b**, **d**, **g**). For the boxplots, the middle line is the median, the lower and upper hinges correspond to the first and third quartiles, and whiskers represent ±1.5 interquartile range (**f**, **h**). Two-tailed Student's $t$-test. **$P < 0.01$. ***$P < 0.001$. ****$P < 0.0001$.

maximizing therapeutic outcomes[49]. Notably, the causes of radioresistance are polymodal and associated with not only oxygen tension, but also other important factors such as cellular energetics, changes in DNA repair, angiogenesis, inflammation, and growth signaling pathways[50,51]. Herein, the photothermal effect from the pHPFON carrier was utilized not only for controllable O$_2$ release but also for cell ablation by carefully adjusting the laser irradiation duration, therefore bridging the gap between PTT and the pHPFON-NO/O$_2$-boosted RT (oxyRT) for significant cancer cell killing. The synergistic effects between PTT and RT were mainly from four perspectives. First, mild hyperthermia arising from PTT could speed up intratumor blood flow to improve tumor oxygenation[12]. Second, the pHPFON-NO/O$_2$, on the one hand, would gradually release NO in response to the acidic TME for inhibition of cell respiration; on the other hand, would on-demand release O$_2$ with local hyperthermia stimuli, providing a "broadening sources of O$_2$ and reducing expenditure" strategy for effective hypoxia attenuation. Third, hyperthermia could effectively inhibit the nonlethal damage repair of ionizing irradiation[52,53], thus potentiating RT damage. Fourth, although hypoxia is the main cause of radioresistance, other factors such as

cellular energetics, inflammation, and growth signaling pathways may also adversely impact RT[51]. The PTT effect could be further increased to kill the tumor residuals whose radioresistant properties were not originated from hypoxia. Taken together, the combination of PTT and RT achieved significantly synergistic effect for complete tumor control.

The safety issue is always a major concern for clinical translation of nanomedicine. According to the US Food and Drug Administration, synthetic amorphous silica is used as food additive and is generally recognized as a safe material. The metabolic profile of MSNs has been extensively explored. Silica nanoparticles, synthesized with different methods and with sizes ranging from 50 to 200 nm, demonstrated predominant renal clearance at the first 24 h while hepatobiliary clearance at later time points[54–56]. Coating MSNs with a polymer layer, such as hyaluronic acid and poly(ethylene glycol), did not significantly impact their metabolism and excretion tendency[56,57]. Therefore, we reasonably speculated that our pHPFON-NO/O$_2$ would be mainly excreted from the body through urine at the early stage and then cleared from liver and spleen at the later stage. The in vivo imaging data in Fig. 6b well demonstrated our hypothesis

by showing gradually decreased accumulation of pHPFON-NO/$O_2$ in liver and spleen after 24 h of injection. Moreover, the complete blood count and blood chemistry analysis found that all tested parameters were in normal physiological ranges after 7 and 14 days of the pHPFON-NO/$O_2$ injection. No apparent acute pathological changes were identified from histological analysis after the injection of pHPFON-NO/$O_2$. Collectively, our pHPFON-NO/$O_2$ silica nanoformulation is safe and holds great potential in clinical translation.

In summary, we rationally designed a unique binary oxygen-economizer pHPFON-NO/$O_2$ to overcome the hypoxia barrier in antitumor RT. By taking full advantage of the facile framework hybridization technique, we granted the pHPFON a powerful "all-in-one" theranostic nanoplatform with intrinsic NIR-II fluorescence imaging/PA imaging/PTT effect and $O_2$ delivery capability. By employing the in situ polymerization method, we addressed the premature leakage issue associated with most mesoporous nanocarriers, thus achieving efficient NO prodrug loading. We demonstrated that the spontaneous NO release from the pHPFON-NO/$O_2$ while penetrating the TME or entering cancer cells could not only reduce intrinsic oxygen consumption by inhibiting cell respiration but also dilate tumor vasculature for enhanced tumor accumulation of the nano-sized drug delivery system. Upon laser irradiation, the accumulated pHPFON-NO/$O_2$ would release $O_2$ to further improve tumor oxygenation. Meanwhile, the photothermal effect could be easily controlled to induce not only mild hyperthermia for $O_2$ release alone, but also a high local temperature for concurrent tumor destruction. This study established a "broadening source of $O_2$ and reducing expenditure" strategy for significant reversal of hypoxia tolerance in oxygen-dependent anti-cancer therapies. It also demonstrated the synergy between temperature-controlled PTT and oxygen-elevated RT for complete tumor response.

## Methods

**Synthesis of SPB-Br**. DPP-Sn (0.85 g, 1 mmol), DPP-Br (0.39 g, 0.5 mmol), fluorine-based monomer 1 (0.29 g, 0.5 mM), bis(triphenylphosphine)palladium(II) dichloride (7.1 mg, 0.01 mmol), and 2,6-di-*tert*-butylphenol (4.2 mg, 0.02 mmol) were mixed in a 10 mL Schlenk flask. Next, 3 mL of toluene and a stirring bar were added. The mixture was subjected to three freeze–pump–thaw cycles before being filled with argon. Then, the flask was heated to 100 °C under vigorous stirring for 6 h. After the reaction, the green solution was dropwisely added to excess methanol to precipitate out solid crude product. The product was collected by filtration and purified by re-dissolving into tetrahydrofuran (THF) and precipitating with excess methanol. After repeating this purification process three times, the final SPB-Br polymer was obtained (1.22 g, yield: 80%).

**Synthesis of SPB-N₃**. SPB-Br (100 mg) was dissolved into a 10 mL mixture of equal volume of THF and dimethylformamide (DMF). Next, sodium azide (2 eq. to bromide group of SPB-Br) was added. The reaction was carried out under vigorous stirring at 25 °C for 48 h. After removing THF, the solution was added dropwise into an excess amount of methanol. The precipitate was obtained by filtration and purified by re-dissolving into 5 mL of chloroform and precipitating with excess methanol. After repeating the purification process three times, the final SP-N₃ was obtained after drying under vacuum for 24 h.

**Synthesis of SPB-PEG₅₀₀₀**. A click reaction was applied to synthesize the SPB-PEG₅₀₀₀. Briefly, SPB-N₃ (10 mg) was dissolved into 5 mL of THF solution. Next, CuBr (2.2 mg, 2 eq. to the azide group of SPB-N₃), *N,N,N′,N″,N‴*-penta-methyldiethylenetriamine (PMDETA, 10 mg), and mPEG₅₀₀₀-alkyne (Mn = 5000, 4 eq. to the azide group of SPB-N₃) were added. The reaction was carried out at room temperature under argon for 48 h. After removing THF, the remaining residue was dissolved into water and dialyzed against deionized water for 3 days. The final SPB-PEG₅₀₀₀ was obtained by freeze-drying.

**Synthesis of SPB silane (SPBS)**. Carboxyl group-based SPB was first synthesized by dissolving SPB-PEG₅₀₀₀ into excess trifluoroacetic acid under stirring for 24 h at room temperature. Then, the SPBS was synthesized by stirring 100 mg of carboxyl group-based SPB, an appropriate amount of *N*-(3-dimethylaminopropyl)-*N′*-ethylcarbodiimide (EDC), an appropriate amount of *N*-hydroxysuccinimide

(NHS), and 5 mg of 3-(triethoxysilyl)propylamine, in 5 mL of DMF for 12 h. The DMF solution was used without further purification for the next step.

**Synthesis of hybridized HPFON**. First, core–shell structured MSN@MON was synthesized according to the chemical homology principle. Briefly, 20 mL of 2 g cetyltrimethylammonium chloride (CTAC) and 0.1 g triethanolamine aqueous solution was stirred for 30 min. After transferring the system to an 80 °C oil bath, 1 mL of TEOS was added and reacted for 1 h. Next, 1 mL of an equal volume mixture of 1,4-bis(triethoxysilyl)benzene (BTEB), PFDES, and SPBS was added and reacted for 4 h. The products were centrifuged and washed with ethanol several times. The CTAC residue was extracted with 30 mL of methanol with NaCl (1 wt %). Second, 5 mL of the above as-synthesized MSN@MON aqueous solution was added to 25 mL ultrapure water. After addition of ammonium hydroxide, the system was transferred to a 95 °C oil bath and reacted for 3 h to etch away the inner MSN core. The resultant hybridized HPFON was centrifuged and washed with water several times. For other formulations, the BTEB was removed or was replaced with TEOS or BTES, respectively.

**In situ polymerization of HPFON (pHPFON)**. The hybridized HPFON (100 mg) and 2,2′-Azobis(2-methylpropionitrile) (AIBN, 5 mg, 0.05 mmol) were dissolved in 2 mL of DMF in a 10 mL Schlenk flask. Then, the mixture was added with freshly distilled isobutyl methacrylate (284.4 mg, 2 mmol), poly(ethylene glycol) methyl ether methacrylate (mPEGMA) (475 mg, 0.5 mmol), and a stirring bar. After deoxygenating by argon blowing for 30 min, the mixture was vigorously stirred for several hours at room temperature to ensure thorough diffusion of polymer monomer and initiator in the silica cavity. Subsequently, the mixture solution was heated up to 65 °C and reacted for 2 h. The reaction mixture was added to excess ethanol, and the precipitate was collected by centrifugation. After that, this ethanol precipitation process was repeated twice to completely remove unconjugated polymers.

**Synthesis of pHPFON-NO/$O_2$**. For NO prodrug loading, the pHPFON and appropriate amount of NO-releasing molecule SNAP were dissolved in 5 mL DMF. Then, 5 mL $H_2O$ was added into the solution under ultrasonication and stirred overnight. The resultant pHPFON-NO nanoparticles were washed by centrifugation and re-dissolved into ultrapure water for further use. For $O_2$ saturation, 1 mL of the pHPFON-NO or pHPFON aqueous solution was placed in an aseptic oxygen chamber for 30 min at an oxygen flow rate of 5 L/min, resulting in pHPFON-NO/$O_2$ or pHPFON-$O_2$, respectively.

**Cumulative NO and $O_2$ release**. The release of NO from pHPFON-NO was evaluated at different pH values (5.5, 6.5, and 7.4). Briefly, pHPFON-NO was suspended in PBS buffer and incubated at 37 °C on a 200 r.p.m. orbital shaker. Fifty microliters of the solution was collected from the suspension at selected time points (0, 0.5, 1, 2, 3, 4, 5, 6, 8, 10, 12, and 24 h), respectively. NO concentration was measured by the Griess reagent system according to the manufacturer's protocol. The release of $O_2$ from pHPFON-$O_2$ was measured by a portable dissolved oxygen meter in deoxygenated water in real time. An 808-nm laser was applied at a dosage of 1 W/cm² for 10 min and the oxygen concentration was recorded every 30 s. To compare, the oxygen release from deoxygenated water upon laser irradiation was monitored. Deoxygenated water with or without addition of pHPFON-$O_2$ were measured in the absence of laser irradiation as additional controls.

**In vitro NO and $O_2$ detection**. The intracellular NO and $O_2$ levels were detected with fluorogenic DAF-FM diacetate and [Ru(dpp)₃]Cl₂ probes, respectively. U87MG cells were incubated overnight at 37 °C under an atmosphere of 1% $O_2$/5% $CO_2$/94% $N_2$. Then, 10 μM DAF-FM diacetate and 10 μg/mL [Ru(dpp)₃]Cl₂ were added and incubated for 30 min. After washing away free fluorescent probes, the cells were incubated with 200 μg/mL of various pHPFON formulations (i.e., pHPFON-NO/$O_2$, pHPFON-NO, pHPFON-$O_2$, and pHPFON) or PBS for 24 h in separate experiments. Subsequently, the cells were subjected to 808-nm laser irradiation at 1 W/cm² for 3 min. To compare, another group of cells were treated with the above nanoparticles or PBS but without receiving the laser irradiation. Finally, the intracellular fluorescent signals were examined with confocal micro-scopy and flow cytometry analysis. For confocal imaging, the cells were fixed with 4 % paraformaldehyde and mounted with mounting medium with DAPI before the examination. For flow cytometry analysis, the cells were trypsinized and collected before the analysis.

**In vitro evaluation of RT**. For MTT studies, U87MG cells were seeded in a 96-well plate at a density of $10^4$ cells per well and incubated overnight at 37 °C under hypoxic atmosphere (1% $O_2$/5% $CO_2$/94% $N_2$). The hypoxic cells were then co-incubated with pHPFON-NO/$O_2$ (200 μg/mL) for 24 h and irradiated with an 808-nm laser (1 W/cm², 3 min). In other groups, the cells were treated with pHPFON-NO alone, pHPFON-$O_2$ + laser, pHPFON + laser, pHPFON alone, or PBS alone. After that, the cells were exposed to X-ray irradiation at a dosage varying from 0, 2, 4, to 6 Gy. After another 48 h of incubation, the cell killing effect was evaluated with the MTT assay. These experiments were repeated with cells that were cultured

under a normoxic environment (21% $O_2$/5% $CO_2$/74% $N_2$) for further comparison. For the colony formation assay, U87MG cells were seeded in a six-well plate. Then, hypoxic U87MG cells were treated with pHPFON-NO/$O_2$ (200 µg/mL) for 24 h under 1% $O_2$ and sequentially irradiated with an 808-nm laser at 1 W/cm$^2$ for 3 min and X-rays (pHPFON-NO/$O_2$ + laser + X-ray) at various dosages (0, 2, 4, 6 Gy). In other groups, the cells received the following treatments: pHPFON-NO + X-ray, pHPFON-$O_2$ + laser + X-ray, pHPFON + laser + X-ray, pHPFON + X-ray, X-ray alone. After these treatments, the cells were incubated for another 24 h under 1% $O_2$ and then transferred to 21% $O_2$ for 14 days of incubation. The as-formed colonies were fixed and stained with Gimesa. The survival fraction was calculated as the ratio of colony numbers that contained more than 50 cells. These treatments were again repeated with cells that were incubated under normoxic environment. For comet assays, cells were seeded in a six-well plate at a density of $10^5$ cells per well. Then, hypoxic cells received the same treatments as those in the colony formation assay. After that, the cells were washed, collected, and fixed in slides for a cell gel electrophoresis assay. In the above experiments, all the nanoparticles were at the same concentration of 200 µg/mL; the laser irradiation was at 808 nm and 1 W/cm$^2$ for 3 min; and the X-ray irradiation was at a dosage of 4 Gy unless otherwise stated.

**In vitro biological effect with the O₂ release**. Hypoxic U87MG cells were treated with pHPFON-$O_2$ (200 µg/mL) for 24 h under 1% $O_2$. Subsequently, the cells were irradiated with an 808-nm laser at 1 W/cm$^2$ for 3 min. For controls, cells were treated with PBS alone, laser irradiation alone, pHPFON-$O_2$ alone, or pHPFON + laser. After washing with PBS three times, the cells were subjected to anti-HIF-1α staining. To further evaluate the radiosensitization of $O_2$ release, the cells were subjected to either a 4-Gy X-ray irradiation or remained in the dark in separate experiments after the above treatments. Finally, intracellular ROS level and DNA damage were evaluated with an $H_2DCFDA$ assay (pre-incubation of 10 µM $H_2DCFDA$ with cells for 20 min before the nanoparticle incubation) and anti-γ-H2Aχ staining through confocal microscopy, respectively.

**In vitro biological effect with the NO release**. Hypoxic U87MG cells were treated with pHPFON-NO (200 µg/mL), pHPFON (200 µg/mL), and PBS for 24 h under 1% $O_2$, respectively. After that, the spent medium was removed, and the cells were washed with PBS for three times. The cellular C*c*O activity (Sigma-Aldrich, CYTOCOX1), cellular ATP concentration (Cayman Chemical, 700410), and cellular oxygen consumption capacity (Cayman Chemical, 600800) were measured according to the vendor's protocols. The cellular mitochondria membrane potential change and cellular HIF-1α level were visualized with JC-1 assays and anti-HIF-1α immunofluorescence staining for confocal microscopy analysis, respectively. In addition, the pHPFON-NO concentration-dependent cellular C*c*O activity was investigated. In brief, hypoxic U87MG cells were treated with pHPFON-NO at different concentrations (0, 25, 50, 100, 200, and 400 µg/mL) for 24 h under 1% $O_2$. After washing with PBS three times, the cells were acquired, and the mitochondria were collected with a mitochondria isolation kit. Then, the C*c*O activity was measured with an assay kit under a microplate reader. Moreover, the impact of NO release on RT was evaluated. Hypoxic U87MG cells were treated with pHPFON-NO (200 µg/mL), pHPFON (200 µg/mL), and PBS for 24 h under 1% $O_2$, respectively. Then, these cells were either treated with 4-Gy X-ray irradiation or kept in the dark. Finally, the intracellular ROS level was detected with an $H_2DCFDA$ assay (pre-incubation of 10 µM $H_2DCFDA$ with cells for 20 min before the nanoparticle incubation), and DNA damage was evaluated with anti-γ-H2Aχ staining through confocal microscopy.

**In vitro synergistic photothermal and RT**. Hypoxic U87MG cells were incubated with pHPFON-NO/$O_2$ (200 µg/mL) for 24 h under 1% $O_2$. Subsequently, the cells were sequentially irradiated with 5 min of laser irradiation (808 nm, 1 W/cm$^2$) and a 4-Gy X-ray (T1). In the control groups, the cells were treated with (T2) pHPFON-NO/$O_2$ + 5-min laser, (T3) pHPFON-NO/$O_2$ + 3-min laser + 4-Gy X-ray, (T4) pHPFON-NO/$O_2$ + 3-min laser, (T5) 4-Gy X-ray, (T6) 5-min laser, (T7) pHPFON-NO/$O_2$, and (T8) PBS. After the treatments, the cells were washed three times with PBS and replenished with fresh medium. After another 48 h of incubation, MTT assays, live/dead cell viability assays, and Annexin V-FITC/PI dual-staining assays were performed to evaluate the cell killing effect.

**In vivo imaging**. For PET imaging, the $^{64}$Cu-labeled pHPFON-NO/$O_2$ was intravenously administrated into U87MG tumor-bearing mice ($n = 3$) at a dosage of 100 µCi. Then, PET images were acquired at 1, 4, 24, and 48 h p.i. Regions of interest (ROIs) were circled and the corresponding radioactivities were quantified on the tumors, muscles, livers, and spleens in the decay-corrected whole-body coronal images. At end of the imaging, the mice were scarified. Major organs were collected and weighted and their radioactivities were measured by gamma counting. For NIR-II and PA imaging, 200 µL of pHPFON-NO/$O_2$ (20 mg/mL) was intravenously administrated into U87MG tumor-bearing mice ($n = 3$). Then, the nanoparticle migration was monitored at selected time points. The tumor accumulation was measured according to the acquired images. For comparison, the tumor accumulation of pHPFON was also monitored with PA imaging after intravenous injection.

**In vivo photothermal effect**. pHPFON-NO/$O_2$ (200 µL of 20 mg/mL in PBS) and PBS were intravenously administrated into U87MG tumor-bearing mice ($n = 3$) in separate groups. After 24 h, an 808-nm laser was irradiated at the tumor areas at 1 W/cm$^2$ for 5 min. An infrared camera was utilized to record thermal images during the irradiation. The temperature of the tumor was measured according to the thermal images.

**In vivo hypoxia evaluation**. The intratumoral oxygen level change was evaluated with Oxy-PA imaging before and after 4 and 24 h of the intravenous injection of pHPFON-NO/$O_2$. Subsequently, an 808-nm laser was irradiated on the tumors at 1 W/cm$^2$ for 3 min. After 30 min, the tumors were subjected to PA imaging to investigate the saturated oxygen level and collected for anti-HIF-1α staining. In control groups, tumors that treated with pHPFON-NO/$O_2$ alone, laser irradiation alone, and PBS alone were also acquired for anti-HIF-1α staining. In addition, to investigate tumor hypoxia alleviation with mere $O_2$ or NO, tumors treated with pHPFON-$O_2$ + Laser or pHPFON-NO were acquired and subjected to anti-HIF-1α staining.

**In vivo tumor inhibition**. U87MG tumor models were established by subcutaneous inoculation of the cells into the right hindlimb of nude mice. When the tumor size reached around 100 mm$^3$, the mice ($n = 5$) were intravenously injected with pHPFON-NO/$O_2$ (200 µL of 20 mg/mL in PBS). After 24 h, tumors were sequentially exposed to 5 min of laser irradiation (808 nm, 1.0 W/cm$^2$) and X-ray irradiation (8 Gy) (recognized as T1). The seven control groups received: (T2) pHPFON-NO/$O_2$ + 5-min laser, (T3) pHPFON-NO/$O_2$ + 3-min laser + X-ray, (T4) pHPFON-NO/$O_2$ + 3-min laser, (T5) X-ray, (T6) 5-min laser, (T7) pHPFON-NO/$O_2$, and (T8) PBS. Tumor sizes (=length × width$^2$/2) and mice body weights were monitored every 3 days. In a separate study, tumors were collected 48 h after the above treatments, and major organs were collected at the end of the therapy for H&E staining and TUNEL assays. In another study, tumors were collected 30 min after the above treatments for anti-γ-H2Aχ staining analysis. Besides, two other control groups, pHPFON-$O_2$ + 3-min laser and pHPFON-NO, were performed and the tumors were acquired for anti-γ-H2Aχ staining analysis for further comparison.

**Reporting summary**. Further information on research design is available in the Nature Research Reporting Summary linked to this article.

## Data availability

The experimental data supporting the findings of this study are available within the article and the Supplementary Information. Extra data are available from the corresponding authors upon reasonable request. Source data are provided with this paper.

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

## Acknowledgements

This research was supported by the intramural research program of the National Institute of Biomedical Imaging and Bioengineering (NIBIB), National Institutes of Health (NIH). We also gratefully acknowledge support from the National Natural Science Foundation of China (NSFC) (Grant #: 21874024).

## Author contributions

W.T., Z.Y., W.F., and X.C. conceived and designed the project. W.F. and Z.Y. synthesized the materials. W.T., W.F., Z.Y., B.S., J.S., J.Z., P.H., M.W., and L.T. performed the material characterizations. W.T., L.D., L.L., J.M., and Y.C. performed the in vitro experiments and analyzed the data. L.H. and S.Z. performed the NIR-II imaging experiment. Z.W. and O.J. performed the PET imaging experiment. W.T., W.F., L.D., and Y.M. performed the other in vivo experiments and analyzed the data. W.T., Z.Y., W.F., P.F., and X.C. co-wrote the paper. All the authors approved the final version.

## Competing interests

The authors declare no competing interests.
