## [Peer Review File · Nature Communications]

REVIEWER COMMENTS

Reviewer #1 (Remarks to the Author):

The authors build on their previous work where they developed different kinds of nanoparticles for theranostic purposes in cancer research. Here, they developed a hybrid semiconducting organosilica-based O₂ nanoeconomizer pHPFON-NO/O₂. The main goal for developing this pHPFON-NO/O₂ nanoplatform was to overcome the hypoxia resistance for antitumor radiotherapy. They aimed to do this by a three-fold system: First, the nanoplatform reduces hypoxia by secreting NO in the tumor, which reduces the oxygen consumption of cancer cells ("reducing expenditure"). Second, the nanoplatform reduces hypoxia by supplying exogenous oxygen ("broadening sources"). Third, the possibility exists to exploit the photothermal effect of the laser irradiation to apply hyperthermia in the tumor. Next to this, they claim that the nanoplatform can be used as a diagnostic tool as well with intrinsic NIR-II fluorescence as with photoacoustic imaging.

The research is well-performed and they provided enough evidence to convince the reader that the nanoplatform can radiosensitize cancer cells and in vivo tumors. The main mechanisms being the release of NO from the pHPFON-NO/O₂ while penetrating the tumor microenvironment (TME), which reduces cellular oxygen consumption and dilates tumor vasculature, and the release of O₂ after laser irradiation to further improve tumor oxygenation. The authors also prove that the photothermal effect of laser irradiation can be controlled and works synergistically with radiotherapy.

Overall, I would like to congratulate the authors on a job well done. The use of NO and adding O₂ to alleviate tumor hypoxia have already been known to radiosensitize tumors for quite a while. Still, the combination of these two modalities in a nanoplatform is what makes this research novel. It is also interesting that there is a possibility to use hyperthermia to further radiosensitize the tumor or kill the cells that are radioresistant by other mechanisms than hypoxia.

However, I think the paper could benefit from revision, primarily targeted at increasing the level of depth in the discussion and more effectively communicating the complexities associated with the data and resulting interpretations. A lot of different concepts are discussed, so I am not entirely convinced that the paper will be accessible to and/or of interest to a broad audience (material scientist, radiobiologist, radiotherapists,...). Below, I have detailed several major suggestions, several minor suggestions and some optional experimental questions.

I praise the authors for an exciting study and encourage them to continue expanding its depth and accessibility.

Major suggestions:

1. Line 47 – 49: Researchers not familiar with the basics of radiotherapy will not understand why hypoxia leads to radioresistance. I suggest the authors explain here the oxygen fixation hypothesis (Brown, Nat Rev Cancer, 2004).
2. Line 49-54: I understand that the authors want to focus on the use of nanoparticles to reshape the hypoxic TME, but in my opinion, the article can benefit from a more elaborate discussion (in introduction or discussion) about radiosensitization by reshaping hypoxia. Normobaric oxygen, Hyperbaric oxygen, nitroimididazoles, hypoxic cytotoxins, hyperthermia,... (Overgaard, J Clin Oncol, 2007)
3. Line 72 – 88: Next to its effects on tumor vessels and mitochondria, NO has a direct radiosensitizing effect on the DNA damage that is induced by radiotherapy (De Ridder et al, 2008; Howard-Flanders et al,1957) and on several immune cells. I suggest elaborating on every effect of NO in the TME.
4. Are there any nanomaterials already used to incorporate NO for radiosensitization? If so, this would also fit in the introduction/discussion.
5. Figure 4 and 5: It is difficult for the reader to see the different effects of pHPFON-NO, pHPFON-O₂ and pHPFON-NO/O₂ on the cells and the general effect on radiotherapy. I would suggest making figure 4 about the effects of the nanoparticles on the cells (fig 4 a and b + fig 5 d,g,b,c,e)

and fig 5 about the radiosensitizing effect and the possible mechanisms (fig 4 c, d,e,f,g + fig 5 i,j). Then you can rewrite the results and discussion part were you discuss first the separate effects and then switch to the radiosensitizing effect with the underlying mechanisms.

6. Line 275-293: The whole part discusses figures that are supplementary. I suppose the authors find the photothermal effect important enough to put in the paper. I would suggest making either a separate figure or adding some of the graphs to the other figures.

Minor suggestions:

1. Line 65-66: The claim that oxygen consumption in cancer cells is high is not entirely in line with the literature. It is still accepted that most tumors have a high glycolytic metabolism and do not use oxidative phosphorylation as a major source of energy.

2. Line 66: Inhibition of oxygen for alleviating hypoxia was first proposed by Secomb et al, 1995.

3. Line 70: There are more articles in literature that used metformin or phenformin and other drugs as hypoxic radiosensitizers: Zanella et al, 2013; de Mey et al, 2018; Ashton et al, 2016; Wheaton et al, 2014.

4. Figure 4: I suppose that the name NIRhmon stands for pHPFON?

5. Line 498: From which vendors are the kits that were used?

6. Figures 4 and 7: The numbering of g1-8 is confusing. I propose to either put the treatments in writing on the graph or to use a different number system and be systematic.

Optional experimental questions:

1. Why only use the U87MG glioblastoma model?

2. Why did you only stain with Hif1 α and no other hypoxia dyes? Hif1 α can also be influenced by a lot of different factors next to hypoxia.

3. Do you have the colony formation assays with multiple radiation doses? Since this is the golden standard, it can be interesting to show the survival curves of the cells.

Reviewer #2 (Remarks to the Author):

The manuscript described a innovative “reducing expenditure of O₂ and broadening sources” strategy significantly alleviated tumor hypoxia in multiple ways, greatly enhanced the therapeutic efficacy of radiation in vitro and in vivo, and demonstrated the synergy between on-demand temperature-controlled photothermal and oxygen-elevated radiotherapy for complete tumor response.

Major concerns:

(1) This manuscript is generally well written and the claims are well demonstrated, especially in oxygen consumption, NO release and synergetic strategy. However, the discussion section was somehow weak. The novelty, significance and necessity of PTT and RT synergy strategy should also be discussed in a great depth.

(2) The combinational therapy appears effective in vitro and in vivo. The in vitro characterization is sufficient, but there are some shortcomings in various of the more correlative in vivo analyses, such as the organizational gamma-H2AX activation, which would be direct evidence to confirm the synergistic effect for pHPFON-NO/O₂-boosted RT.

(3) The manuscript presents very interesting pre-clinical data and the therapeutical results are really excellent. However, what is the final distribution of the pHPFON-NO/O₂? How could pHPFON-NO/O₂ be metabolized, and passed out of the body? These points are really important and should be study.

Minor concerns:

(1) All the figure legends do not state whether any of the experiments have been repeated, and whether the results match. This information is essential, especially in cell assays. Besides, the statistical methods should be provided in the figure legends.

(2) The bar of Fig. 4a (pHPFON-NO-Laser) was missed.

(3) The Fig. 4c and 4d, “+Laser”, the Laser irradiation time and power were not specified.

- (4) The pictures of cell colony assay (corresponding to Fig. 4e) should be provided.
- (5) The line 286-287, For "The synergy between PTT and oxyRT was verified by the calculation of $f_{additive} = (f_{PTT} \times f_{oxyRT}) < f_{PTT+oxyRT}$."
- i) Please provide the theoretical basis of calculation formula: $f_{additive} = (f_{PTT} \times f_{oxyRT})$. Why should it be calculated like this ?
- ii) Actually, $f_{additive} = (f_{PTT} \times f_{oxyRT}) = 38.8 \% \times 68.3 \% = 26.5 \% > f_{PTT+oxyRT} = 20.4 \%$. Please explain it.
- (6) Since the authors have multi-mode imaging information (PET imaging, NIR-II imaging and PA imaging), it would be valuable to calculate and compare the accumulation percentage of pHPFON-NO/O₂ in tumor tissue under different imaging conditions.
- (7) The data upon sensitization enhancement ratio of PTT-boosted RT in animal models should be provided.
- (8) It would be much clearer if the figure legends communicated the RT dose given and the timing when samples were harvested for analysis.

Reviewer #3 (Remarks to the Author):

In this manuscript, the authors prepared a type of hollow nanoparticles based nanotheranostics enabling tumor acidity/NIR light responsive release of NO and O₂ for effective tumor hypoxia relief and enhanced cancer radiotherapy. However, I think that these currently presented results are not solid enough to support the synergistic effect of NO and O₂ in tumor hypoxia attenuation as claimed in this manuscript. Therefore, I think that this work needs more improvements before it could be considered for publication.

1. To confirm the synergistic effects of the as-prepared nanotheranostics in tumor hypoxia relief and radiotherapy, the effects of these control groups enabling release of bare NO or O₂ on tumor hypoxia relief and radiotherapy treatments should be carefully studied and compared with the one enabling simultaneous release of NO and O₂ in parallel.
2. In Figure 3i, it was shown that the oxygen concentration of the deoxygenated water was ~0. However, based on our previous experience, the oxygen concentration of the deoxygenated water was ~5-6 mg/L. Please double check.
3. In Figure 4f, the background fluorescence signals in different groups were different. Please double check.
4. In Figure 5g, the fluorescence signals of JC-1 aggregate and JC-1 monomer are suggested to be provided.
5. In Figure 5i, the background green fluorescence signal of NIRmon-O₂ treatments group was quite higher than other groups. Please double check.

Author's Response to Reviewer #1 (In Blue)

The authors build on their previous work where they developed different kinds of nanoparticles for theranostic purposes in cancer research. Here, they developed a hybrid semiconducting organosilica-based O₂ nanoeconomizer pHPFON-NO/O₂. The main goal for developing this pHPFON-NO/O₂ nanoplatform was to overcome the hypoxia resistance for antitumor radiotherapy. They aimed to do this by a three-fold system: First, the nanoplatform reduces hypoxia by secreting NO in the tumor, which reduces the oxygen consumption of cancer cells (“reducing expenditure”). Second, the nanoplatform reduces hypoxia by supplying exogenous oxygen (“broadening sources”). Third, the possibility exists to exploit the photothermal effect of the laser irradiation to apply hyperthermia in the tumor. Next to this, they claim that the nanoplatform can be used as a diagnostic tool as well with intrinsic NIR-II fluorescence as with photoacoustic imaging.

The research is well-performed and they provided enough evidence to convince the reader that the nanoplatform can radiosensitize cancer cells and *in vivo* tumors. The main mechanisms being the release of NO from the pHPFON-NO/O₂ while penetrating the tumor microenvironment (TME), which reduces cellular oxygen consumption and dilates tumor vasculature, and the release of O₂ after laser irradiation to further improve tumor oxygenation. The authors also prove that the photothermal effect of laser irradiation can be controlled and works synergistically with radiotherapy.

Overall, I would like to congratulate the authors on a job well done. The use of NO and adding O₂ to alleviate tumor hypoxia have already been known to radiosensitize tumors for quite a while. Still, the combination of these two modalities in a nanoplatform is what makes this research novel. It is also interesting that there is a possibility to use hyperthermia to further radiosensitize the tumor or kill the cells that are radioresistant by other mechanisms than hypoxia.

However, I think the paper could benefit from revision, primarily targeted at increasing the level of depth in the discussion and more effectively communicating the complexities associated with the data and resulting interpretations. A lot of different concepts are discussed, so I am not entirely convinced that the paper will be accessible to and/or of interest to a broad audience (material scientist, radiobiologist, radiotherapists,...). Below, I have detailed several major suggestions, several minor suggestions and some optional experimental questions.

I praise the authors for an exciting study and encourage them to continue expanding its depth and accessibility.

RE: Thanks very much for your positive and constructive comments. Per your comments, we have added in-depth discussion on the oxygen fixation hypothesis, clinical attempts in reshaping hypoxia for reversal of radiation resistance, radiosensitizing effect of NO in the tumor microenvironment (TME), nanodelivery systems of NO. In addition, Figs. 4 & 5 were also re-arranged to first explore the *in vitro* release and mechanism of the radiosensitizing effect of NO and O₂ and then examine the efficacy of hypoxia attenuation-sensitized radiotherapy (RT) as well as synergistic photothermally-boosted RT. We have carefully revised the manuscript according to

your other comments and suggestions. All changes are highlighted in red. Point-by-point responses to your comments are listed below.

Major suggestions:

1. Line 47 – 49: Researchers not familiar with the basics of radiotherapy will not understand why hypoxia leads to radioresistance. I suggest the authors explain here the oxygen fixation hypothesis (Brown, *Nat Rev Cancer*, 2004).

RE: We thank the reviewer for the constructive suggestion. The presence of oxygen in tumors has substantial impact on treatment outcome; relative to anoxic regions, well-oxygenated cells respond better to radiotherapy by a factor of 2.5–3. The oxygen effect is most commonly explained by the “oxygen fixation hypothesis”, which postulates that radical-induced DNA damage (DNA•) can be permanently fixed by molecular oxygen to generate DNA-OO•, rendering DNA damage irreparable. However, the DNA• radical enters into a competition for reduction under hypoxic condition, primarily by –SH-containing compounds that can restore the DNA to its original form. Therefore, DNA damage is less in the absence of oxygen in radiotherapy.

We employed this oxygen fixation hypothesis to describe the possible mechanisms of O₂ release for RT sensitization in the re-arranged Fig. 4j (Fig. 5h in the original manuscript). In the revised manuscript, we have also added the following discussion to the introduction part.

“In RT, ionizing radiation damages cells by producing a radical on DNA (DNA•). According to the oxygen fixation hypothesis, the DNA radical can be further oxidized by molecular O₂ to generate DNA-OO•, thus inducing the damage fixation and DNA double strand breaks. Of note, the radical can be competitively reduced at the same time, especially under hypoxic condition, by thiol-containing compounds to restore the DNA to its original form, resulting in less DNA damage.”

2. Brown, J.M. & Wilson, W.R. Exploiting tumour hypoxia in cancer treatment. *Nat. Rev. Cancer* 4, 437-447 (2004).

2. Line 49-54: I understand that the authors want to focus on the use of nanoparticles to reshape the hypoxic TME, but in my opinion, the article can benefit from a more elaborate discussion (in introduction or discussion) about radiosensitization by reshaping hypoxia. Normobaric oxygen, Hyperbaric oxygen, nitroimidazoles, hypoxic cytotoxins, hyperthermia,... (Overgaard, *J Clin Oncol*, 2007)

RE: We thank the reviewer for the comments. The methods of clinically attempting to overcome hypoxic radioresistance mainly focus on increasing oxygen delivery through blood circulation system, mimicking the oxygen effect in RT with nitroimidazole, and employing hypoxia-activatable drugs to directly destroy hypoxic cells. Although hypoxic modification still has limited impact on general clinical practice, ample data exist to support a high level of evidence for the benefit of hypoxic modification in radiotherapy.

The following discussion has been added to the introduction and discussion parts.

“Various methods of modifying hypoxic radioresistance have been explored in clinical trials, such as increasing oxygen delivery through the blood with hyperbaric oxygen (HBO), normobaric oxygen/carbogen breathing, nicotinamide, blood transfusion, erythropoietin, or a combination of them; mimicking the oxygen effect on fixation of radiation-induced DNA damage in the radiochemical process with nitroimidazoles; destroying hypoxic cells, rather than sensitizing them, with hypoxic cytotoxins; and having a more direct radiation target in the cells with high linear energy transfer irradiation.⁴ But hypoxic modification in the routine clinical situation remains inconclusive and very limited, partly because the above strategies are small and underpowered, or due to the involvement of techniques that are difficult to be practiced routinely (e.g., HBO, etc.).^{4”}

4. Overgaard, J. Hypoxic radiosensitization: adored and ignored. *J. Clin. Oncol.* **25**, 4066-4074 (2007).

3. Line 72 – 88: Next to its effects on tumor vessels and mitochondria, NO has a direct radiosensitizing effect on the DNA damage that is induced by radiotherapy (De Ridder et al, 2008; Howard-Flanders et al, 1957) and on several immune cells. I suggest elaborating on every effect of NO in the TME.

RE: Many thanks for the suggestion. NO is an efficient hypoxic radiosensitizer, as it may mimic the effects of oxygen on fixation of radiation-induced DNA damage, but the required levels cannot be obtained *in vivo* because of vasoactive complications. Strategies with endogenous production of NO at tumor site may overcome this issue. For example, isoform of NO-synthase (iNOS), activated by pro-inflammatory cytokines, was demonstrated to be capable of radiosensitizing tumor cells through endogenous production of NO, at non-toxic extracellular concentrations. It has also been confirmed that the radiosensitizing effect is transcriptionally controlled by hypoxia and by NF- κ B. In addition, tumor-associated immune cells (e.g., macrophages, T/NK-cells, etc.) may contribute to the iNOS-mediated radiosensitization by the generation of pro-inflammatory cytokines and NO, which may diffuse towards bystander tumor cells.

Per your suggestion, we have added the following discussion in the manuscript.

“For example, Howard-Flanders demonstrated NO as an effective radiosensitizer as early as 1957 on hypoxic bacteria.³¹ They proposed that the primary mechanism of NO-based radiosensitization was to fix radiation-induced DNA damage and mimic the oxygen effects on DNA lesions.³¹ Yet, it required a high level of NO concentration which may not be obtained *in vivo* due to its vasoactive complications. An alternative mechanism might be the interaction of NO with the oxygen-binding sites in mitochondria,³² leading to inhibition of cell respiration and conservation of physiological oxygen for sensitizing RT.³³ Of note, De Ridder *et al.* first reported that NO can be endogenously generated through inducible isoform of NOS (iNOS) for radiosensitization.³⁴ On this basis, proinflammatory tumor infiltrates, for example, activated macrophages and T/NK-cells, can sensitize hypoxic tumors to RT through iNOS-dependent pathways by production of pro-inflammatory mediators and NO.^{35, 36} But the percentage of the tumor-associated immune cells

varies in different tumor types and it may need to combine with immunostimulators for enough NO production.”

31. Howardflanders, P. Effect of nitric oxide on the radiosensitivity of bacteria. *Nature* **180**, 1191-1192 (1957).
32. Mason, M.G., Nicholls, P., Wilson, M.T. & Cooper, C.E. Nitric oxide inhibition of respiration involves both competitive (heme) and noncompetitive (copper) binding to cytochrome *c* oxidase. *Proc. Natl. Acad. Sci. U.S.A.* **103**, 708-713 (2006).
33. Mitchell, J.B. et al. Radiation sensitisation by nitric oxide releasing agents. *Br. J. Cancer Suppl.* **27**, 181-184 (1996).
34. Janssens, M.Y., Van den Berge, D.L., Verovski, V.N., Monsaert, C. & Storme, G.A. Activation of inducible nitric oxide synthase results in nitric oxide-mediated radiosensitization of hypoxic EMT-6 tumor cells. *Cancer Res.* **58**, 5646-5648 (1998).
35. Cavaillon, J.M. Cytokines and macrophages. *Biomed. Pharmacother.* **48**, 445-453 (1994).
36. Matsuura, M., Saito, S., Hirai, Y. & Okamura, H. A pathway through interferon-gamma is the main pathway for induction of nitric oxide upon stimulation with bacterial lipopolysaccharide in mouse peritoneal cells. *Eur. J. Biochem.* **270**, 4016-4025 (2003).

4. Are there any nanomaterials already used to incorporate NO for radiosensitization? If so, this would also fit in the introduction/discussion.

RE: Thanks for the question and suggestion. Many nanomaterials have been used to deliver NO for reversing multi-drug resistance, enhancing passive cancer targeting, and sensitizing photodynamic therapy. But few of them focused on NO delivery for radiosensitization. For example, Gao et al. used PLGA-*b*-PEG nanocarrier to deliver DM1-NO conjugate, in which DM1 inhibited microtubule polymerization and enriched cells at the G2/M phase while the NO under radiation formed highly toxic radicals such as peroxy nitrates to suppress tumor growth (*ACS Nano*, **2020**, *14*, 1468). The PLGA-*b*-PEG system had high drug loading but failed to demonstrate diagnostic capacity. Fan et al. designed an upconversion nanotheranostic system, PEG-USMSs-SNO, by engineering upconversion nanoparticle (UCNP) with NO releasing molecule-grafted mesoporous silica. It sensitively responded to X-ray irradiation to release NO for on-demand hypoxic radiosensitization besides upconversion luminescent imaging through UCNPs both *in vitro* and *in vivo* (*Angew Chem. Int. Ed. Engl.*, **2015**, *54*, 14026). But the NO releasing molecules were loaded to the mesoporous channels of the mesoporous silica without adding capping agents on the pores, which raised concerns in premature release during the blood circulation. To overcome the above shortcomings, in this manuscript, we proposed an *in situ* polymerized, hollow-structured, and semiconducting polymer brush (SPB) and fluorocarbon (FC) chain co-hybridized organosilica nanoplatfrom (pHPFON). The *in situ* polymerization provided opportunities for efficient drug loading with strong hydrophobic-hydrophobic interactions and thus well addressed the premature release problem. Meanwhile, the framework hybridization of SPB allowed the nanocarrier for excellent near-infrared (NIR) II fluorescence and photoacoustic contrast for imaging purposes.

We have added the following discussion in the revised manuscript:

“Several nanocarriers have been reported to incorporate NO releasing molecules for radiosensitization, such as poly(lactide-co-glycolic)-block-poly(ethylene glycol) (PLGA-b-PEG) nanoparticles³⁷ and upconversion nanoparticle-engineered mesoporous silica (USMS) core-shell structures.³⁸ However, these carriers are either lack of diagnostic functionality or involved with potential pre-mature drug release issues. More improvements can be made in the design of the nanocarriers.”

37. Gao, S. et al. Nanoparticles encapsulating nitrosylated maytansine to enhance radiation therapy. *ACS Nano* **14**, 1468-1481 (2020).

38. Fan, W. et al. X-ray radiation-controlled NO-release for on-demand depth-independent hypoxic radiosensitization. *Angew Chem. Int. Ed. Engl.* **54**, 14026-14030 (2015).

5. Figure 4 and 5: It is difficult for the reader to see the different effects of pHPFON-NO, pHPFON-O₂ and pHPFON-NO/O₂ on the cells and the general effect on radiotherapy. I would suggest making figure 4 about the effects of the nanoparticles on the cells (fig 4 a and b + fig 5 d,g,b,c,e) and fig 5 about the radiosensitizing effect and the possible mechanisms (fig 4 c, d,e,f,g + fig 5 i,j). Then you can rewrite the results and discussion part were you discuss first the separate effects and then switch to the radiosensitizing effect with the underlying mechanisms.

RE: We appreciate the reviewer’s comment and suggestion. In the original manuscript, Figs. 4 a and b were about *in vitro* NO and O₂ release; Figs. 5 d, g, b, c, e explored the possible mechanisms behind the NO-sensitized RT, which is to inhibit mitochondrial respiration, down-regulate HIF-1 α expression, and spare physiological O₂ for DNA damage fixation; Figs. 5 i and j investigated the mechanism behind the O₂-sensitized RT; Figs. 4 c, d, e described the radiosensitizing effect of different nanoparticles; Figs. 4 f and g further evaluated the RT effect with comet assay by examining RT-induced oxidative damage to DNA.

To make it easier to follow, we have rearranged Figs. 4 & 5. The revised Fig. 4 describes the *in vitro* NO and/or O₂ release and the possible mechanisms behind their radiosensitizing effect (Fig 4 a and b + Fig 5 in the original manuscript). The revised Fig. 5 is about the *in vitro* therapeutic effects of the nanoparticles on RT, which includes Fig. 4 c,d,e,f,g in the original manuscript to compare the efficacy among RT alone, NO-sensitized RT, O₂-sensitized RT, and NO/O₂ dual-sensitized RT (oxyRT). In addition, to better illustrate the synergy between photothermal therapy (PTT) and oxyRT, we have moved the MTT assay, live/dead double staining analysis, and flow cytometry analysis results of the combination PTT/oxyRT treatment from supplementary to the revised Figs. 5f and g. The results and discussion part were re-organized accordingly.

6. Line 275-293: The whole part discusses figures that are supplementary. I suppose the authors find the photothermal effect important enough to put in the paper. I would suggest making either a separate figure or adding some of the graphs to the other figures.

RE: We thank the reviewer for the suggestion. We have moved the MTT assay, live/dead double staining, and flow cytometry analysis results of the photothermal effect-boosted oxyRT to the revised Figs. 5 f, g, h, i, accordingly.

The updated Figs. 4 & 5 are shown as below.

Figure 4. *In vitro* programmable release and radiosensitizing effect of NO and O₂. (a) Confocal images of hypoxic U87MG cells treated with different pHPFON formulations for 24 h, with or without subsequent 808-nm laser irradiation (1 W/cm², 3 min). Green, DAF-FM (NO indicator). Red, [Ru(dpp)₃]Cl₂ (hypoxia indicator). Blue, DAPI. Scale bar, 20 μm. Experiments were performed three times with similar results. (b) Flow cytometry analysis of hypoxic U87MG cells receiving the same treatments in (a). Experiments were performed twice with similar results. (c) Schematic illustration of the NO delivery-based "reducing expenditure" oxygenation strategy for boosted RT. The low-pH-induced NO release would inhibit mitochondrial respiration, down-regulate HIF-1α expression, and boost RT efficacy. (d) Relative activity of cytochrome c oxidase

(CcO) after incubating hypoxic U87MG cells with pHPFON-NO at different concentrations for 24 h. n = 4 biologically independent samples. Data are presented as mean \pm s.d. (e)-(i) Effect of cell respiration inhibition by the pHPFON-NO. n = 4 biologically independent samples per group (e, g). Experiments were performed three times with similar results (f, h, i). (e) Relative CcO activity, (f) JC-1 assay, (g) relative ATP contents, (h) oxygen consumption capacity, and (i) HIF-1 α expression after co-incubation of hypoxic U87MG cells with pHPFON-NO, pHPFON, or PBS overnight. (j) Schematic illustration of the O₂ delivery-based “broadening sources” oxygenation strategy for advanced RT. The laser-activatable O₂ release would down-regulate HIF-1 α expression and augment X-ray-induced oxidative DNA damage. (k) Anti-HIF-1 α staining in hypoxic U87MG cells after different treatments. Green, HIF-1 α . Red, tubulin. Experiments were performed three times with similar results. (l) Evaluation of intracellular ROS generation and DNA damage with H₂DCFDA assay and anti- γ -H2A γ staining after different treatments. Green, DCF or γ -H2A γ . Blue, DAPI. Scale bar: 20 μ m. Experiments were performed three times with similar results. (+) stands for 808-nm laser irradiation at 1 W/cm² for 3 min applied after 24 h of incubation with nanoparticles. (#) stands for 4-Gy X-ray irradiation following the laser irradiation, if applicable. Two-tailed Student’s *t*-test. *** *P* < 0.001.

Figure 5. *In vitro* radiotherapy. (a) – (e) NO and/or O₂-boosted radiotherapy. (a) & (b) Cell viabilities of (a) normoxic (21% O₂) and (b) hypoxic (1% O₂) U87MG cells subjected to different nanoparticle treatments, following by an X-ray irradiation at various doses (0, 2, 4, 6 Gy). In the groups with laser irradiation, the laser (808 nm) was applied after 24 h of incubation at a dosage of 1 W/cm² for 3 min. n = 5 biologically independent samples per group. (c) Survival fraction determined by colony formation assays in both normoxic and hypoxic U87MG cells after different treatments. n = 3 biologically independent samples per group. (d) Fluorescent DNA-stained images by comet assays in hypoxic U87MG cells after different treatments. Scale bar: 50 μm. Experiments were performed three times with similar results. (e) Quantification of DNA damage (n = 6) according to the images in (d). Groups g1-g8: g1, pHPFON-NO/O₂ + Laser + X-ray; g2, pHPFON-

NO + X-ray; g3, pHPFON-O₂ + Laser + X-ray; g4, pHPFON + Laser + X-ray; g5, pHPFON + X-ray; g6, X-ray; g7, pHPFON-NO/O₂ + Laser; g8, PBS. Laser (808 nm) was applied after 24 h of incubation with nanoparticles at 1 W/cm² for 3 min. X-ray was applied after the laser irradiation at a dose of 4 Gy. (f)-(i) *In vitro* synergistic photothermal and radiotherapy. (f) MTT assays (n = 5), (g) Live and dead assays (n = 3, with similar results), (h) Flow cytometry analysis (n = 2, with similar results), and (i) Quantitative analysis according to (h) on cells after different treatments. Green, Calcein AM, live cells. Red, Eth-1, dead cells. Scale bar, 100 μm. Groups T1-T8: T1, pHPFON-NO/O₂(++)(#); T2, pHPFON-NO/O₂(++); T3, pHPFON-NO/O₂(+)(#); T4, pHPFON-NO/O₂(+); T5, (#); T6, (++); T7, pHPFON-NO/O₂; and T8, PBS. (++) stands for 808-nm laser irradiation at 1 W/cm² for 5 min. (+) stands for 808-nm laser irradiation at 1 W/cm² for 3 min. (#) stands for a 4-Gy X-ray irradiation. Laser was applied after 24 h of incubation with nanoparticles and X-ray was applied after the laser irradiation, if applicable. Data are presented as mean ± s.d. Two-tailed Student's *t*-test. * *P* < 0.05. ** *P* < 0.01. *** *P* < 0.001. *n.s.*, not significant.

Minor suggestions:

1. Line 65-66: The claim that oxygen consumption in cancer cells is high is not entirely in line with the literature. It is still accepted that most tumors have a high glycolytic metabolism and do not use oxidative phosphorylation as a major source of energy.

RE: We thank the reviewer for the critique. We agree that most tumors have high glycolytic metabolism to acquire energy for growth and proliferation (“Warburg effect”). However, significantly elevated oxidative phosphorylation (OXPHOS) rates still appear in many malignant cell lines during tumorigenesis, development, and metastasis. For example, the cell respiration rate of MCF-7 human breast carcinoma was measured to be 7 nmol O₂/min mg cell protein while that of MCF10A human breast epithelial cells was 2 nmol O₂/min mg cell protein (*Int. J. Biochem. Cell Biol.* **2010**, 42, 1744. *Biochim. Biophys. Acta.* **2012**, 1817, 1597.); the rates of HTB-126 human breast carcinoma and HTB-125 human non-cancer breast cells were 28.5 and 12 nmol O₂/min mg cell protein, respectively (*J. Bioenerg. Biomembr.* **2010**, 42, 55); and the rates of HCC4017 human non-small-cell lung cancer and WI-38 human embryonic lung fibroblasts were 12.4 and 0.5-0.75 nmol O₂/min mg cell protein, respectively (*PLoS ONE* **2013**, 8, 63402. *Free Radic. Biol. Med.* **2011**, 15, 700. *Biotechnol. Bioeng.* **2004**, 86, 775). After reviewing the causes of tumor hypoxia, we noticed that respiration is the major way for living cells to consume O₂. Therefore, we believe that blocking the endogenous O₂ depletion would be a promising and meaningful approach to modulate the hypoxia status.

To emphasize the importance of both the “Warburg effect” and cell respiration, we have added a discussion on the “Warburg effect” and re-written the sentence as follows.

“Despite that cancer cells acquire energy primarily through aerobic glycolytic metabolism, or the “Warburg effect”, mitochondrial respiration is not diminished in cancer cells.¹⁵ Instead, it plays an important role in tumor development and progression, which is evidenced by significantly elevated mitochondrial respiration rates in many cancer cell lines.^{16, 17”}

15. Koppenol, W.H., Bounds, P.L. & Dang, C.V. Otto Warburg's contributions to current concepts of cancer metabolism. *Nat. Rev. Cancer* **11**, 325-337 (2011).

16. Moreno-Sanchez, R. et al. Who controls the ATP supply in cancer cells? Biochemistry lessons to understand cancer energy metabolism. *Int. J. Biochem. Cell Biol.* **50**, 10-23 (2014).
17. Moreno-Sanchez, R., Rodriguez-Enriquez, S., Marin-Hernandez, A. & Saavedra, E. Energy metabolism in tumor cells. *FEBS J.* **274**, 1393-1418 (2007).

2. Line 66: Inhibition of oxygen for alleviating hypoxia was first proposed by Secomb et al, 1995.

RE: We appreciate the reviewer's comment. Secomb proposed that the effects of blood flow rate, blood oxygen content, and oxygen consumption on hypoxic fraction can be simulated theoretically. They analyzed a region whose microvascular geometry was derived from observations of a transplanted mammary adenocarcinoma (R3230AC) in a rat dorsal skin flap preparation. They found that hypoxia was abolished by a reduction in consumption rate of at least 30%, relative to control. These results suggested that reducing oxygen consumption rate may be an alternative method for effective hypoxia attenuation.

We have added the reference and the following discussion in the revised manuscript.

“The concept of inhibiting oxygen consumption for hypoxia attenuation was first proposed by Secomb et al. in 1995.¹⁸ Since then, several chemodrugs such as metformin,¹⁹⁻²² phenformin,²¹ and atovaquone²³⁻²⁵ have been applied to disturb mitochondrial respiration for improved tumor oxygenation.”

18. Secomb, T.W., Hsu, R., Ong, E.T., Gross, J.F. & Dewhirst, M.W. Analysis of the effects of oxygen supply and demand on hypoxic fraction in tumors. *Acta. Oncol.* **34**, 313-316 (1995).
19. Song, X.J. et al. Liposomes co-loaded with metformin and chlorin e6 modulate tumor hypoxia during enhanced photodynamic therapy. *Nano Res.* **10**, 1200-1212 (2017).
20. Zannella, V.E. et al. Reprogramming metabolism with metformin improves tumor oxygenation and radiotherapy response. *Clin. Cancer Res.* **19**, 6741-6750 (2013).
21. de Mey, S. et al. Antidiabetic biguanides radiosensitize hypoxic colorectal cancer cells through a decrease in oxygen consumption. *Front. Pharmacol.* **9**, 1073 (2018).
22. Wheaton, W.W. et al. Metformin inhibits mitochondrial complex I of cancer cells to reduce tumorigenesis. *Elife* **3**, 02242 (2014).
23. Xia, D.L. et al. Overcoming hypoxia by multistage nanoparticle delivery system to inhibit mitochondrial respiration for photodynamic therapy. *Adv. Funct. Mater.* **29**, 1807294 (2019).
24. Wang, D. et al. Inhibiting tumor oxygen metabolism and simultaneously generating oxygen by intelligent upconversion nanotherapeutics for enhanced photodynamic therapy. *Biomaterials* **251**, 120088 (2020).
25. Ashton, T.M. et al. The anti-malarial atovaquone increases radiosensitivity by alleviating tumour hypoxia. *Nat. Commun.* **7**, 12308 (2016).

3. Line 70: There are more articles in literature that used metformin or phenformin and other drugs

as hypoxic radiosensitizers: Zanella et al, 2013; de Mey et al, 2018; Ashton et al, 2016; Wheaton et al, 2014.

RE: We have added these references in the revised manuscript.

4. Figure 4: I suppose that the name NIRhmon stands for pHPFON?

RE: Thanks for pointing out the typo. We have now revised the name of NIRhmon to pHPFON in that figure.

5. Line 498: From which vendors are the kits that were used?

RE: The cellular cytochrome *c* oxidase (CcO) activity assay kit was purchased from Sigma-Aldrich (catalog #: CYTOCOX1). The ATP detection assay kit and oxygen consumption rate assay kit were purchased from Cayman Chemical company (Item No. 700410, 600800). The vendor information was provided in the materials part in the Supporting Information. We have now stated it in the Methods section along with the experimental procedure in our revised manuscript.

6. Figures 4 and 7: The numbering of g1-8 is confusing. I propose to either put the treatments in writing on the graph or to use a different number system and be systematic.

RE: We thank the reviewer for the suggestion. To make it clear, we have used a different numbering system (T1-8) for experimental groups in the combinational PTT and oxyRT treatment studies both *in vitro* and *in vivo*. The numbering of g1-8 remains unchanged to represent experimental groups in the NO and/or O₂-sensitized RT studies. The numbering of T1-8 has been updated in the context, figures, and figure captions in the revised manuscript.

Optional experimental questions:

1. Why only use the U87MG glioblastoma model?

RE: This study mainly focused on tumor hypoxia modulation for radiosensitization, thus we chose a hypoxic tumor model. Glioblastoma (GBM) is one of the most hypoxic tumor types. As shown in a review paper titled “Exploiting tumor hypoxia in cancer treatment” (*Nat. Rev. Cancer*, **2004**, 437), the authors summarized a table (listed below) on oxygenation of different types of tumors and the surrounding normal tissue. According to their summary, the medium tumor pO₂ of glioblastoma is 4.9 – 5.6 mmHg, whereas those of head and neck carcinoma, lung cancer, and breast cancer are 12.2 – 14.7, 7.5, and 10.0 mmHg, respectively. Moreover, many evidences show that GBMs are intrinsically radioresistant (*PNAS*, **1998**, 95, 14453; *Oncotarget* **2017** 8, 100931; *PLOS ONE* **2019**, 14, 0215714). In particular, U87MG is more radioresistant than U251, which is partly attributable to more cycling U251 cells found in G2/M, the most radiosensitive cell stage, while more U87MG cells are found in S and G1, the more radioresistant cell stage (*J. Radiat. Res.*,

2010, 51, 393). All these make U87MG glioblastoma model a valuable candidate for our studies on hypoxia attenuation and radiosensitization.

Table 1 | **Oxygenation of tumours and the surrounding normal tissue**

Tumour type	Median tumour pO ₂ * (number of patients)	Median normal pO ₂ * (number of patients)	References
Glioblastoma	4.9 (10)	ND	128
	5.6 (14)	ND	129
Head and neck carcinoma	12.2 (30)	40.0 (14)	130
	14.7 (23)	43.8 (30)	131
	14.6 (65)	51.2 (65)	132
Lung cancer	7.5 (17)	38.5 (17)	Q. Le (personal communication)
Breast cancer	10.0 (15)	ND	133
Pancreatic cancer	2.7 (7)	51.6 (7)	134
Cervical cancer	5.0 (8)	51 (8)	135
	5.0 (74)	ND	136
	3 (86)	ND	137
Prostate cancer	2.4 (59)	30.0 (59)	138
Soft-tissue sarcoma	6.2 (34)	ND	139
	18 (22)	ND	140

*pO₂ measured in mmHg. Measurements were made using a commercially available oxygen electrode (the 'Eppendorf' electrode). The values shown are the median of the median values for each patient. ND, not determined; pO₂, oxygen partial pressure.

In addition, we have ample experience in U87MG model, especially in tumor hypoxia modulation. In our previous study, we observed that the U87MG would develop enough hypoxia when its volume reaches around 60 mm³ through PA/US imaging and immunofluorescence staining analysis (*ACS Nano*, 2018, 12, 1580). That study demonstrated that tumor-specific delivery of O₂-saturated perfluoropentane could achieve hypoxic radiosensitization (*ACS Nano*, 2018, 12, 1580). In another study, we used semiconducting polymer-stabilized perfluorocarbon nanodroplets for *in situ* O₂ delivery, which successfully relieved hypoxia and boosted photodynamic therapy against U87MG tumors (*ACS Nano*, 2018, 12, 2610). In this project, we proposed to explore new tumor hypoxia alleviation strategies. To better evaluate the radiosensitization effect, we continued to use U87MG model since we can learn from our previous data, and compare those with what we observed in this study. As U87MG model is one of the most hypoxic and radioresistant tumor models, it well represents the tumor models we need in this project. We only studied the U87MG glioblastoma model at the current stage. We are very enthusiastic to broaden the application of the pHPFON-NO/O₂ to other tumor models and will explore its theranostic benefits to other tumor models in our subsequent studies in the near future.

2. Why did you only stain with HIF1 α and no other hypoxia dyes? HIF1 α can also be influenced by a lot of different factors next to hypoxia.

RE: Thank you for the question. We agree that HIF1 α can be influenced by a lot of different factors next to hypoxia. It would better if the hypoxia level could be confirmed by other probes, in addition to HIF1 α . At the cellular level, we have used a hypoxia indicator, Ru(dpp)₃(PF₆)₂, to monitor the

oxygen and nitrogen release of the nanoparticles in Figs. 4a & b, and Supplementary Figure 10. These results were in good agreement with HIF1 α immunofluorescence staining results in the revised Fig4 i & k. (re-arranged from Figs. 5g & i), suggesting that HIF1 α staining was able to successfully reflect the cellular hypoxia in this study.

For *in vivo* study, one of the most classical hypoxia markers is pimonidazole (*Nat. Nanotechnol.* **2019**, *14*, 1160. *ACS Nano* **2019**, *13*, 1784). It forms covalent bonds with cellular macromolecules at oxygen levels below 1.3% (*Int. J. Cancer* **1995**, *61*, 567.) and visualizes poorly oxygenated regions in histological sections from tumors (*Cancer Sci.* **2009**, *100*, 1366). For analysis of tumor hypoxia, pimonidazole solution (Hypoxyprobe, 60 mg/kg) needs to be intravenously injected at least 1 h before sacrificing the animals. Hypoxia was then assessed in frozen tissue sections by immunostaining of pimonidazole using an anti-Hypoxyprobe antibody. Due to COVID-19 restrictions, we do not currently have access to live animals. We hope the anti-HIF1 α immunofluorescence staining on tumor session in Fig. 7e together with the PA imaging of tumor oxygenation in Fig. 7c could convince the reviewer and readers on the tumor hypoxia alleviation effect of our proposed nanosystem. Hypoxia detection with pimonidazole will be further explored in our subsequent studies in the near future.

3. Do you have the colony formation assays with multiple radiation doses? Since this is the golden standard, it can be interesting to show the survival curves of the cells.

RE: Thanks very much for the comments. Yes, we performed the colony formation assays with multiple radiation doses. We agree that the colony formation assay is the golden standard to evaluate therapeutic effect in RT. The results of dose-dependent radiosensitizing effect of NO and O₂ by colony formation assays have now been added to Supplementary Fig. 18 in the revised manuscript.

Supplementary Figure 18. Survival fraction determined by colony formation assays in both (a) normoxic and (b) hypoxic U87MG cells after different treatments. Laser was applied after 24 h of nanoparticle incubation at a dosage of 1 W/cm² for 3 min. X-ray was applied after the laser irradiation (if applicable) at various doses (0, 2, 4, and 6 Gy). n = 3 biologically independent

samples per group. Data are presented as mean \pm s.d. Two-tailed Student's *t*-test. * $P < 0.05$. ** $P < 0.01$.

Author's Response to Reviewer #2 (In Blue)

The manuscript described an innovative “reducing expenditure of O₂ and broadening sources” strategy significantly alleviated tumor hypoxia in multiple ways, greatly enhanced the therapeutic efficacy of radiation *in vitro* and *in vivo*, and demonstrated the synergy between on-demand temperature-controlled photothermal and oxygen-elevated radiotherapy for complete tumor response.

RE: Thank you very much for your positive and insightful comments. We have carefully revised the manuscript according to your comments and suggestions. All the changes are highlighted in red. Point-by-point responses to your comments are listed below.

Major concerns:

(1) This manuscript is generally well written and the claims are well demonstrated, especially in oxygen consumption, NO release and synergetic strategy. However, the discussion section was somehow weak. The novelty, significance and necessity of PTT and RT synergy strategy should also be discussed in a great depth.

RE: We appreciate the reviewer for the comment. Our study used mild PTT for on-demand release of a large amount of O₂ to the tumor site, which effectively alleviated tumor hypoxia for boosted RT. Moreover, the PTT effect could be precisely controlled to further increase temperature, exerting tumoricidal effects on the remaining cancer cells whose radioresistant properties were caused by other factors.

It is well-known that the oxygen-deficient TME severely decreases the cancer cells' sensitivity to X-ray irradiation which makes RT ineffective in treating hypoxic solid tumors. Hyperthermia arising from PTT has been observed to be able to speed up the intratumoral blood flow for improved tumor oxygenation, which counteracts the hypoxia-induced radioresistance for enhanced RT efficacy. In addition, hyperthermia can effectively suppress the nonlethal damage repair of X-ray irradiation, which gives rise to remarkable synergistic PTT/RT effects via the enhancement of PTT on RT. Increased intratumoral blood flow would not be sufficient for tumor oxygenation. Our study used hyperthermia arising from PTT to control O₂ release from the pHPFON-NO/O₂, providing a “broadening source of O₂” strategy to significantly boost RT. Moreover, the causes of radioresistance are polymodal and associated with not only oxygen tension, but also other important factors such as cellular energetics, changes in DNA repair, angiogenesis, inflammation, and growth signaling pathways. Therefore, it is hard to completely eradicate radioresistant tumors by modifying the hypoxic TME alone. To overcome this problem, the temperature of the PTT

effect can be further increased to exert tumoricidal effects on the residual cells for complete tumor response, making it possible for the combination of RT and PTT.

The following in depth discussion on the PTT and RT synergy strategy has been added in the revised manuscript.

“The synergistic effects between PTT and RT were mainly from four perspectives. First, mild hyperthermia arising from PTT could speed up intratumor blood flow to improve tumor oxygenation.¹² Second, the pHPFON-NO/O₂, on the one hand, would gradually release NO in response to the acidic TME for inhibition of cell respiration; on the other hand, would on-demand release O₂ with local hyperthermia stimuli, providing a novel “broadening sources of O₂ and reducing expenditure” strategy for effective hypoxia attenuation. Third, hyperthermia could effectively inhibit the nonlethal damage repair of ionizing irradiation,^{52, 53} thus potentiating RT damage. Fourth, although hypoxia is the main cause of radioresistance, other factors such as cellular energetics, inflammation, and growth signaling pathways may also adversely impact RT.⁴¹ The PTT effect could be further increased to kill the tumor residuals whose radioresistant properties were not originated from hypoxia. Taken together, the combination of PTT and RT achieved significantly synergistic effect for complete tumor control.”

12. Tang, W. et al. Organic semiconducting photoacoustic nanodroplets for laser-activatable ultrasound imaging and combinational cancer therapy. *ACS Nano* **12**, 2610-2622 (2018).
41. Buckley, A.M., Lynam-Lennon, N., O'Neill, H. & O'Sullivan, J. Targeting hallmarks of cancer to enhance radiosensitivity in gastrointestinal cancers. *Nat. Rev. Gastroenterol. Hepatol.* **17**, 298-313 (2020).
52. Elming, P.B. et al. Hyperthermia: The optimal treatment to overcome radiation resistant hypoxia. *Cancers (Basel)* **11** (2019).
53. Zolzer, F., Streffer, C. & Pelzer, T. Induction of quiescent S-phase cells by irradiation and/or hyperthermia. II. Correlation with colony forming ability. *Int. J. Radiat. Biol.* **63**, 77-82 (1993).

(2) The combinational therapy appears effective *in vitro* and *in vivo*. The *in vitro* characterization is sufficient, but there are some shortcomings in various of the more correlative *in vivo* analyses, such as the organizational gamma-H₂AX activation, which would be direct evidence to confirm the synergistic effect for pHPFON-NO/O₂-boosted RT.

RE: We thank the reviewer for the comments and suggestions. We have added anti- γ -H2A χ immunofluorescence staining on tumor tissues after different treatments in the therapy studies to the revised Supplementary Figure 24. An increasing positive fluorescence signal was observed in the RT (T5), oxyRT (T3) and PTT+oxyRT (T1) group, indicating enhanced oxidative DNA damage and RT efficacy. In T3 group, pHPFON-NO/O₂ induced mild hyperthermia with laser irradiation to release O₂ for RT sensitization. Whereas in T1 group, the PTT induced a relatively high temperature to induce both O₂ release and photoablation. The results well confirmed the on-demand synergistic photothermally-boosted RT. More discussion and experimental details can be found in the revised manuscript and supplementary information.

Supplementary Figure 24. Anti- γ -H2A γ staining on tumor samples acquired at 30 min after different treatments in Fig. 7g. Green, γ -H2A γ . Blue, DAPI. Scale bar, 100 μ m. Experiments were performed three times with similar results.

(3) The manuscript presents very interesting pre-clinical data and the therapeutic results are really excellent. However, what is the final distribution of the pHPFON-NO/O₂? How could pHPFON-NO/O₂ be metabolized, and passed out of the body? These points are really important and should be study.

RE: We appreciate the reviewer for the comments and questions. The final distribution of pHPFON-NO/O₂ was determined by measuring the *ex vivo* radioactivities of the ⁶⁴Cu-labeled nanoparticles at 48 h post-injection (intravenously), showing an accumulation rate of 5.84, 4.03, 4.78, 12.53, 6.42, and 4.48 %ID/g in tumor, heart, lung, liver, kidneys, and spleen, respectively (Supplementary Fig. 21).

Silica nanoparticles are usually passed out of the body through urine at the early stage and through liver and spleen at the later stage. For example, in Waegeneers et al.'s study (*Toxicol. Rep.* **2018**, 5, 632), a single intravenous injection of NM-200 silica nanoparticle was applied at a dose of 20 mg/kg_{bw}, followed by autopsy after 6 and 24 h. They found the main organs where silicon accumulated were liver and spleen. The silicon concentration significantly decreased in spleen between 6 and 24 h. In liver the tendency was the same but the effect was not significant. This could be due to clearance of the spleen to the liver *via* the splenic vein, while liver clearance takes more time due to hepatic processing and biliary excretion. Within the first 24 h, silica was mainly excreted through urine. In another example, Moghaddam *et al.* compared *in vivo* clearance of three similar sized silica nanoparticles (i.e., Stober 100, Meso 100, and Disulfide Hollow 100) that synthesized with different methods (*J. Control Release* **2019**, 311-312, 1-15). Particles at the dose of 25 mg/kg_{bw} were tail vein injected to immunocompetent CD-1 female mice. After 24 h, renal excretion of Disulfide Hollow 100 nanoparticles was *ca.* 25.9 % while this value was *ca.* 11.6 % and 21.7 % for Stober 100 and Meso 100 particles, respectively. After 7 days, all three

nanoparticles accumulated more in the liver and spleen than in the lung and kidneys. Taken together, the dominant renal clearance was found at the early stage, while hepatobiliary clearance was critical for the excretion of degradation products at the later stage. For this study, we are again very sorry that the metabolic profile of pHPFON-NO/O₂ was not obtained since we have very limited access to live animals due to the COVID-19 restrictions. We have performed multi-modal imaging of pHPFON-NO/O₂ and hope these data are helpful. According to the PET imaging results in Fig. 6a & b, the liver and spleen accumulation of pHPFON-NO/O₂ gradually decreased over time, indicating the clearance from liver and spleen.

Minor concerns:

(1) All the figure legends do not state whether any of the experiments have been repeated, and whether the results match. This information is essential, especially in cell assays. Besides, the statistical methods should be provided in the figure legends.

RE: Thank you for the important suggestions. We have added the information in figure legends in the revised manuscript and supplementary information.

(2) The bar of Fig. 4a (pHPFON-NO-Laser) was missed.

RE: Thank you for the comment. We have added the scale bar to the figure.

(3) The Fig. 4c and 4d, “+Laser”, the Laser irradiation time and power were not specified.

RE: We thank the reviewer for the comment. We have specified the laser dose (808 nm, 1 W/cm², 3 min) in the revised figure caption.

(4) The pictures of cell colony assay (corresponding to Fig. 4e) should be provided.

RE: Thank you for the suggestion. We have examined the survival fractions of U87MG cells by the colony formation assay after different nanoparticle treatments (i.e., pHPFON-NO/O₂ + Laser, pHPFON-NO, pHPFON-O₂ + Laser, pHPFON + Laser, pHPFON, PBS) plus various doses of X-ray irradiation (0, 2, 4, 6 Gy) under both normoxic and hypoxic conditions. The survival fractions were shown in the re-arranged Fig. 5e and Supplementary Fig. 18 (n = 3 biologically independent samples per group). Colony formation assay is the gold standard to determine cell reproductive death after treatment with ionizing radiation. We recorded the numbers of colony formation in each of the 48 experimental conditions to quantify the therapeutic efficacy, but didn't take pictures. To visualize X-ray irradiation-induced DNA damage, comet assays (in the re-arranged Fig. 5d) and anti- γ -H2A χ staining analysis (re-arranged Fig. 4l, Supplementary Figures. 14 & 17) on cells after different RT treatments have been performed. We hope these results can give the reviewer and audience some clues and conclusions on the boosted RT efficacy with the pHPFON-NO/O₂. We are so sorry we cannot re-perform colony assay for the picture acquisition purposes during

the pandemic. Pictures will be taken together with the colony formation counting in our future studies if applicable. Thank you.

(5) The line 286-287, For “The synergy between PTT and oxyRT was verified by the calculation of $f_{\text{additive}} = (f_{\text{PTT}} \times f_{\text{oxyRT}}) < f_{\text{PTT+oxyRT}}$.”

i) Please provide the theoretical basis of calculation formula: $f_{\text{additive}} = (f_{\text{PTT}} \times f_{\text{oxyRT}})$. Why should it be calculated like this?

ii) Actually, $f_{\text{additive}} = (f_{\text{PTT}} \times f_{\text{oxyRT}}) = 38.8 \% \times 68.3 \% = 26.5 \% > f_{\text{PTT+oxyRT}} = 20.4 \%$. Please explain it.

RE: We are grateful to the reviewer’s careful critiques. We had a typo in the calculation equation. The $f_{\text{additive}} = (f_{\text{PTT}} \times f_{\text{oxyRT}})$ should be greater than, instead of less than, $f_{\text{PTT+oxyRT}}$.

According to previous publications (*Proc. Nat. Acad. Sci U.S.A.* **1975**, 72, 937–940. *ACS Nano* **2009**, 3, 2919–2926. *ACS Nano* **2016**, 10, 11027–11036.), the predicted additive survival for combined exposure (in the absence of synergistic interaction) should be a multiplication of the survival rate of each treatment modality, whereas the measured survival for combined exposure should be lower than the predicted one if synergistic effects exist. For example, Hahn et al. (*Proc. Nat. Acad. Sci U.S.A.* **1975**, 72, 937–940) demonstrated synergism between hyperthermia (42–43°C) and adriamycin (or bleomycin) in mammalian cell inactivation in thermochemotherapy. In their experiment, exposure to 30 µg/mL of bleomycin at 37° permitted 40% of the cell population to maintain their reproductive integrity. A 1 h exposure to 43° in the absence of bleomycin could reduce survival by approximately 50 %. Hence, the predicted survival for combined exposure (in the absence of synergistic interaction) should be about 20%. In fact, the measured survival was 4×10^{-4} , lower by a factor of 500. For another example, Li et al. (*ACS Nano* **2016**, 10, 11027–11036.) investigated the synergistic between CO gas therapy and PTT with m-PB-CO/PEG NPs. When HeLa cells were treated with m-PB-PEG NPs (100 ppm), laser irradiation at 0.3 W/cm² had no effect on the survival rate in the absence of hyperthermia, while that at 0.8 W/cm² resulted in 30.2 % cells killed because of the thermal effect. If the cells were treated with m-PB-CO/PEG NPs (100 ppm), 0.3 W/cm² showed 25.6 % cell apoptosis from CO toxicity, and 0.8 W/cm² caused the viability to drop to 51.4 %. The predicted additive survival rate was calculated to be $(1 - 30.2\%) \times (1 - 25.6\%) = 51.93\%$, which is $> 51.4\%$ from the observed combinational CO and PTT treatment.

In this manuscript, the predicted additive survival rate $f_{\text{additive}} = (f_{\text{PTT}} \times f_{\text{oxyRT}}) = 38.8 \% \times 68.3 \% = 26.5 \%$, which is greater than the measured survival rate ($f_{\text{PTT+oxyRT}}$) of 20.4 % in the combination PTT and oxyRT treatment. Therefore, the synergy between PTT and oxyRT was verified by the calculation of $f_{\text{additive}} = (f_{\text{PTT}} \times f_{\text{oxyRT}}) > f_{\text{PTT+oxyRT}}$.

We have corrected the equation to $f_{\text{additive}} = (f_{\text{PTT}} \times f_{\text{oxyRT}}) > f_{\text{PTT+oxyRT}}$ and added the above reference in the revised manuscript.

49. Hahn, G.M., Braun, J. & Har-Kedar, I. Thermochemotherapy: synergism between hyperthermia (42-43 degrees) and adriamycin (of bleomycin) in mammalian cell inactivation. *Proc. Natl. Acad. Sci. U.S.A.* **72**, 937-940 (1975).

50. Park, H. et al. Multifunctional nanoparticles for combined doxorubicin and photothermal treatments. *ACS Nano* 3, 2919-2926 (2009).
51. Li, W.P. et al. Controllable CO release following near-infrared light-induced cleavage of iron carbonyl derivatized Prussian Blue nanoparticles for CO-assisted synergistic treatment. *ACS Nano* 10, 11027-11036 (2016).

(6) Since the authors have multi-mode imaging information (PET imaging, NIR-II imaging and PA imaging), it would be valuable to calculate and compare the accumulation percentage of pHPFON-NO/O₂ in tumor tissue under different imaging conditions.

RE: Thank you for the important suggestion. According to Fig. 6, we have calculated the accumulation percentage of pHPFON-NO/O₂ in tumor tissue under PET imaging and NIR-II imaging. But for PA imaging, the tumor was imaged slice by slice rather than the whole tissue, thus making it hard to give an accurate quantitative evaluation on the accumulation percentage. PET imaging has unlimited penetration. Although both NIR-II fluorescence imaging and PA imaging have relatively high penetration, the later one has high spatial resolution but the former one doesn't. All the three imaging modalities have high sensitivity and can compensate for respective inherent drawbacks. The tumor accumulation percentage of pHPFON-NO/O₂ under PET and NIR-II imaging were calculated and added to Supplementary Figure 22, which demonstrated similar tumor uptake rate at the same time point.

Supplementary Figure 22. Comparison of tumor accumulation of pHPFON-NO/O₂ under PET and NIR-II imaging based on Fig. 6 a and c. n = 3 biologically independent animals. Data are presented as mean ± s.d.

(7) The data upon sensitization enhancement ratio of PTT-boosted RT in animal models should be provided.

RE: We appreciate the reviewer's suggestion. Herein, by adjusting irradiation time, the PTT effect could be precisely controlled to either a mild hyperthermia temperature for O₂ release (oxyRT treatment group), or a relatively high temperature to concurrently release O₂ and exert tumoricidal effects (oxyRT + PTT treatment group). According to the tumor growth curves (Fig. 7g), the tumor inhibition rates (Fig. 7h) of RT, oxyRT, and oxyRT + PTT groups were 55.9%, 70.8%, and 100% at day 18 after the treatments, respectively. Therefore, the *in vivo* sensitization enhancement ratio of PTT-boosted RT was calculated to be $100\% \div 55.9\% = 1.79$ in comparison with RT, or $100\% \div 70.8\% = 1.41$ compared with oxyRT. We have added a statement on this conclusion in the revised manuscript as follows.

“Therefore, the sensitization enhancement ratio of the combination group (Group T1) was calculated to be 1.41 and 1.79 over the oxyRT and RT treatment, respectively.”

(8) It would be much clearer if the figure legends communicated the RT dose given and the timing when samples were harvested for analysis.

RE: Many thanks for the suggestion. We have specified the RT doses and the time point of sample acquisition in the figure legends in the revised manuscript and Supplementary Information.

Author's Response to Reviewer #3 (In Blue)

In this manuscript, the authors prepared a type of hollow nanoparticles based nanotheranostics enabling tumor acidity/NIR light responsive release of NO and O₂ for effective tumor hypoxia relief and enhanced cancer radiotherapy. However, I think that these currently presented results are not solid enough to support the synergistic effect of NO and O₂ in tumor hypoxia attenuation as claimed in this manuscript. Therefore, I think that this work needs more improvements before it could be considered for publication.

RE: Thank you very much for your careful and constructive comments. We have carefully revised the manuscript according to your comments and suggestions. More evidences on the tumor hypoxia attenuation and radiotherapeutic efficacy of bare NO or O₂-sensitized RT treatments have been added. All changes are highlighted in red. Point-by-point responses to your comments are listed below.

1. To confirm the synergistic effects of the as-prepared nanotheranostics in tumor hypoxia relief and radiotherapy, the effects of these control groups enabling release of bare NO or O₂ on tumor

hypoxia relief and radiotherapy treatments should be carefully studied and compared with the one enabling simultaneous release of NO and O₂ in parallel.

RE: Many thanks for the critique and suggestion. Unfortunately we were unable to monitor tumor growth curves of bare NO or O₂-sensitized RT, since we have very limited access to live animals due to the COVID-19 restrictions. We therefore investigated tumor hypoxia alleviation and RT efficacy with tissue samples we acquired before. To evaluate the hypoxia relief, anti-HIF-1 α staining was performed on tumor tissues treated with bare NO or O₂ release. The radiotherapeutic effect was investigated with anti- γ -H₂A χ immunofluorescence staining. A much lower extent of positive HIF-1 α and higher level of γ -H₂A χ fluorescence signals were found in the bare NO or O₂ release group than the simultaneous release group, suggesting the most effective hypoxia alleviation and radiosensitization in the programmable NO/O₂ release group.

We have added the results to Supplementary Figures 26 and 27. More discussion and experiment details can be found in the revised manuscript and supplementary information.

Supplementary Figure 26. Anti-HIF-1 α staining on tumor samples acquired at 30 min after different treatments. (+) stands for 808-nm laser irradiation at 1 W/cm² for 3 min at 24 h p.i. Yellow, HIF-1 α . Blue, DAPI. Scale bar, 20 μ m. Experiments were performed three times with similar results.

Supplementary Figure 27. Anti- γ -H2A γ staining on tumor samples acquired at 30 min after different treatments. (+) stands for 808-nm laser irradiation at 1 W/cm² for 3 min at 24 h p.i. (#) stands for X-ray irradiation following the laser irradiation (if applicable). Green, γ -H2A γ . Blue, DAPI. Scale bar, 100 μ m. Experiments were performed three times with similar results.

2. In Figure 3i, it was shown that the oxygen concentration of the deoxygenated water was \sim 0. However, based on our previous experience, the oxygen concentration of the deoxygenated water was \sim 5-6 mg/L. Please double check.

RE: Many thanks for the critique and suggestion. We have double checked the results and confirmed that the concentration was \sim 0. The different measurements in oxygen concentration of deoxygenated water might result from the use of oxygen probes from different manufactures. The one we used was MW600 PRO dissolved oxygen meter (Milwaukee Instruments, Inc., NC, USA). We deoxygenated water by bubbling it with either nitrogen or argon for 20 min. Five independent repetitive experiments were performed. It was confirmed the oxygen concentration of the deoxygenated water under our experimental condition is \sim 0.

3. In Figure 4f, the background fluorescence signals in different groups were different. Please double check.

RE: We thank the reviewer for the suggestion. We double checked the background fluorescence signals and replaced the high background fluorescence images with more representative ones. The updated images are shown as follows.

Figure 5d. Fluorescent DNA-stained images by comet assays in hypoxic U87MG cells after different treatments. Scale bar: 50 μ m. Experiments were performed three times with similar results.

4. In Figure 5g, the fluorescence signals of JC-1 aggregate and JC-1 monomer are suggested to be provided.

RE: We appreciate the reviewer for the suggestion. The fluorescence signals of JC-1 in the figure has been analyzed and a bar graph has been added to the Supplementary Fig. 11b.

Supplementary Figure 11. JC-1 assays of hypoxic U87MG cells after 24 h of co-incubation with PBS, pHPFON, or pHPFON-NO. Green, JC-1 monomers, low mitochondrial membrane potential. Red, JC-1 aggregates, high mitochondrial membrane potential. Scale bar, 20 μ m. Experiments were performed three times with similar results. (b) The ratio of red to green fluorescence in (a). The results are expressed as the mean \pm SD (n = 6) in each independent experiment. Two-tailed Student's *t*-test. *** $p < 0.001$.

5. In Figure 5i, the background green fluorescence signal of NIRmon-O₂ treatments group was quite higher than other groups. Please double check.

RE: Thank you very much for your suggestion. The Fig 5i in the original manuscript has now been rearranged to revised Fig. 4k. In that figure, we evaluated the HIF-1 α expression after different treatments in hypoxic U87MG cells. According to our experimental data, pHPFON-O₂ (NIRhmon-O₂ is a typo) didn't induce a dramatic decrease in HIF-1 α fluorescence intensities until a laser irradiation applied, which should be attributed to the strong interactions between the adsorbed O₂ and the incorporated perfluorocarbon chains in the nanoparticle's framework. We re-analyzed the data and replaced the figure with a more representative one. The updated results are shown as follows.

Figure 4k. Anti-HIF-1 α staining in hypoxic U87MG cells after different treatments. Green, HIF-1 α . Red, tubulin. Experiments were performed three times with similar results.

REVIEWERS' COMMENTS

Reviewer #2 (Remarks to the Author):

The authors have clarified the most important issues such as the in-depth discussion of the novelty, significance and necessity of PTT and RT synergy strategy, organizational gamma-H2AX activation, experimental repeats, and other minor issues. There are some issues, which should also be addressed.

Major issues

The authors tried to explain the metabolic process of pHPFON-NO/O₂, and listed a lot of relevant literature to speculate on the metabolic process of pHPFON-NO/O₂. However, it was worth noted that pHPFON-NO/O₂ was obvious different from mesoporous silica NPs (MSN), because the pHPFON had been etched from MSN@MON (90oC, NH₃.H₂O, 3h, Fig. 1a), the pHPFON-NO/O₂ were more like a hybrid semiconducting organosilica “polymers” rather than “mesoporous SiO₂ NPs” (the listed literature refers to silica NPs). Due to COVID-19, the authors had very limited access to animal experiments. In discussion part, the authors could reasonably speculate and explain the metabolic process of pHPFON-NO/O₂ by referring other studies. On the other hand, it is difficult for researchers (especially Interdisciplinary readers) to grasp the novelty and significance of the article by writing the experimental Results part and Discussion part together. Therefore, I suggested that the authors could separate the Results part and the Discussion part into separate paragraphs.

Minor issues:

1. Figure 1a, etch conditions of MSN@MON was 90 oC/NH₃.H₂O, which was not consistent with the conditions of synthesis of hybridized HPFON (95 oC). 95 oC or 90 oC? Minor change to clarify that.
2. TUNEL staining method was missing in the manuscript.
3. In Supplementary Information, Supplementary Figure 3, Experiments were performed “threw” times with similar results. Misspelled word “three” .

Reviewer #3 (Remarks to the Author):

After last round revision, the quality of this manuscript has been significantly improved. So, I would like to recommend it to be accepted for publication as it is.

Author's Response to Reviewer #2 (In Blue):

The authors have clarified the most important issues such as the in-depth discussion of the novelty, significance and necessity of PTT and RT synergy strategy, organizational gamma-H₂AX activation, experimental repeats, and other minor issues. There are some issues, which should also be addressed.

Major issues:

The authors tried to explain the metabolic process of pHPFON-NO/O₂, and listed a lot of relevant literature to speculate on the metabolic process of pHPFON-NO/O₂. However, it was worth noted that pHPFON-NO/O₂ was obvious different from mesoporous silica NPs (MSN), because the pHPFON had been etched from MSN@MON (90°C, NH₃•H₂O, 3h, Fig. 1a), the pHPFON-NO/O₂ were more like a hybrid semiconducting organosilica “polymers” rather than “mesoporous SiO₂ NPs” (the listed literature refers to silica NPs). Due to COVID-19, the authors had very limited access to animal experiments. In discussion part, the authors could reasonably speculate and explain the metabolic process of pHPFON-NO/O₂ by referring other studies.

RE: We greatly thank the reviewer for the comments and suggestion. We speculate that the pHPFON-NO/O₂ was mainly cleared from the body through urine at early stage while through liver and spleen at the later stage.

According to our previous literature research, a single-dose intravenous injection of NM-200 silica nanoparticle into mice (20 mg/kg_{bw}) (*Toxicol. Rep.* **2018**, 5, 632) demonstrated significantly decreased silicon concentration in spleen between 6 and 24 h. In liver the tendency was the same, but the effect was not significant. Within the first 24 h, silicon was mainly excreted through urine. For another example, Moghaddam *et al.* compared *in vivo* clearance of three similar-sized silica nanoparticles (i.e., Stober 100, Meso 100, and Disulfide Hollow 100) that synthesized with different methods (*J. Control Release* **2019**, 311-312, 1-15). They found predominant renal clearance at the early stage, with excretion rates ranging from 11.6 % - 25.9 % after 24 hr. After 7 days, all three nanoparticles accumulated in the liver and spleen more than lung and kidney, suggesting hepatobiliary clearance was predominant at the later stage.

We agree with the reviewer that the pHPFON-NO/O₂ were more like a hybrid semiconducting organosilica “polymers” rather than mesoporous SiO₂ nanoparticles we discussed above. In our recent publication, we have investigated the metabolism and excretion of HMSeN@HOMV nanoformulation in BALB/c mice (*Nat. Biomed. Eng.* **2020**, 4, 1102–1116). The HMSeN@HOMV was prepared by coating hyaluronate-modified bacterial outer membrane vesicles onto hollow mesoporous silica nanoparticles. It was observed that at the early stage (from 2 h to 5 d), the degradation products of the HMSeN@HOMV nanoformulation were mainly excreted from the body through urine, indicating dominant renal clearance at the early stage. The renal clearance peaked at day 1 and then gradually decreased. Excretion of the degradation products in faeces began to increase after 5 d of injection, indicating the beginning of the hepatobiliary clearance. The hepatobiliary clearance reached a peak at day 7 after injection and then gradually decreased.

These results showed that renal clearance had a dominant role in the excretion of degradation products of the HMSeN@HOMV nanoformulation, and hepatobiliary clearance was critical for the excretion of degradation products at the later stage. Therefore, coating mesoporous silica nanoparticle with a polymeric layer would not significantly change its clearance profile.

We speculate that our pHPFON nanopatform, showing the same polymer-shell and organosilica-core structure with the HMSeN@HOMV, would share the similar metabolic profile. We expect dominant renal clearance at the early stage while hepatobiliary clearance at the later stage for the pHPFON-NO/O₂ nanoformulation. Moreover, the PET imaging results in Fig. 6a & b showed gradually decreased accumulation of pHPFON-NO/O₂ in liver and spleen over time, further verifying the clearance from liver and spleen at later stage.

We have now added the following discussion to the revised manuscript.

The safety issue is always a major concern for clinical translation of nanomedicine. According to the US Food and Drug Administration, synthetic amorphous silica is used as food additive and is generally recognized as a safe material. The metabolic profile of mesoporous silica nanoparticles has been extensively explored. Silica nanoparticles, synthesized with different methods and with sizes ranging from 50 to 200 nm, demonstrated predominant renal clearance at the first 24 hrs while hepatobiliary clearance at later time points.⁵⁴⁻⁵⁶ Coating mesoporous silica nanoparticles with a polymer layer, such as hyaluronic acid and poly(ethylene glycol), didn't significantly impact their metabolism and excretion tendency.^{56, 57} Therefore, we reasonably speculated that our pHPFON-NO/O₂ would be mainly excreted from the body through urine at the early stage and then cleared from liver and spleen at the later stage. The *in vivo* imaging data in **Fig. 6b** well demonstrated our hypothesis by showing gradually decreased accumulation of pHPFON-NO/O₂ in liver and spleen after 24 h of injection. Moreover, the complete blood count and blood chemistry analysis found that all tested parameters were in normal physiological ranges after 7 and 14 days of the pHPFON-NO/O₂ injection. No apparent acute pathological changes were identified from histological analysis after the injection of pHPFON-NO/O₂. Collectively, our pHPFON-NO/O₂ silica nanoformulation is safe and holds great potential in clinical translation.

54. Waegeneers, N., Brasseur, A., Doren, E.V., Heyden, S.V., Serreyn, P.J., Pussemier, L., Mast, J., Schneider, Y.J., Ruttens, A. & Roels, S. Short-term biodistribution and clearance of intravenously administered silica nanoparticles. *Toxicol. Rep.* **5**, 632-638 (2018).

55. Moghaddam, S.P.H., Mohammadpour, R., Ghandehari, H., In vitro and in vivo evaluation of degradation, toxicity, biodistribution, and clearance of silica nanoparticles as a function of size, porosity, density, and composition. *J. Control Release* **311-312**, 1-15 (2019).

56. Li, L., Zou, J., Dai, Y., Fan, W., Niu, G., Yang, Z. & Chen, X., Burst release of encapsulated annexin A5 in tumours boosts cytotoxic T-cell responses by blocking the phagocytosis of apoptotic cells. *Nat. Biomed. Eng.* **4**, 1102-1116 (2020).

57. Dogra, P. et al. Establishing the effects of mesoporous silica nanoparticle properties on in vivo disposition using imaging-based pharmacokinetics. *Nat. Commun.* **9**, 4551 (2018).

On the other hand, it is difficult for researchers (especially Interdisciplinary readers) to grasp the novelty and significance of the article by writing the experimental Results part and Discussion part together. Therefore, I suggested that the authors could separate the Results part and the Discussion part into separate paragraphs.

RE: Many thanks for the constructive suggestion. We have now separated the Results and Discussion section into respective sections. The Results section mainly focuses on the description and interpretation of experimental data. The Discussion section mainly discusses two topics: one is the novelty, significance, and necessity of PTT and RT synergy strategy; the other is the metabolic process and biosafety of the pHPFON nanoplatform. A summary paragraph is included as well.

Minor issues:

1. Figure 1a, etch conditions of MSN@MON was 90 °C/NH₃•H₂O, which was not consistent with the conditions of synthesis of hybridized HPFON (95 °C). 95 °C or 90 °C? Minor change to clarify that.

RE: Thanks very much for your question. The reaction temperature of the etching process was 95 °C. We have corrected the reaction temperature from 90 to 95 °C in the revised Figure 1a.

2. TUNEL staining method was missing in the manuscript.

RE: We appreciate the reviewer for the suggestion. We have now added the TUNEL staining protocol to the revised Supporting Information.

3. In Supplementary Information, Supplementary Figure 3, Experiments were performed “threw” times with similar results. Misspelled word “three”.

RE: Thanks for pointing out the typo. We have now revised the word spell.